

# Potential of carbon uptake and local aerosol production in boreal and hemi-boreal ecosystems across Finland and in Estonia

Piaopiao Ke[1], Anna Lintunen[1,2], Pasi Kolari[1], Annalea Lohila[1,3], Santeri Tuovinen[1], Janne Lampilahti[1], Roseline Thakur[1], Maija Peltola[1], Otso Peräkylä[1], Tuomo Nieminen[1], Ekaterina Ezhova[1], Mari Pihlatie[4,5], Asta Laasonen[1], Markku Koskinen[4,5], Helena Rautakoski[3], Laura Heimsch[3], Tom Kokkonen[1], Aki Vähä[1], Ivan Mammarella[1], Steffen Noe[6], Jaana Bäck[2], Veli-Matti Kerminen[1], Markku Kulmala[1]

[1] Institute for Atmospheric and Earth System Research (INAR) / Physics, Faculty of Science, University of Helsinki, Helsinki, 00014, Finland
[2] Institute for Atmospheric and Earth System Research (INAR) / Forest Sciences, Faculty of Agriculture and Forestry, University of Helsinki, Helsinki, 00014, Finland
[3] Finnish Meteorological Institute, Finland
[4] Department of Agricultural Sciences, Faculty of Agriculture and Forestry, University of Helsinki, Helsinki 00790, Finland
[5] Department of Agricultural Sciences, Faculty of Agriculture and Forestry, University of Helsinki, Helsinki 00790, Finland
[6] Institute of Forestry and Engineering, Estonian University of Life Sciences, 51006 Tartu, Estonia

Correspondence: Markku.Kulmala (markku.kulmala@helsinki.fi) and Piaopiao Ke (piaopiao.ke@helsinki.fi)



**Abstract:** Continental ecosystems play an important role in carbon dioxide ($CO_2$) uptake and aerosol production, which helps to mitigate climate change. The concept of 'CarbonSink+ potential' enables a direct comparison of $CO_2$ uptake and local aerosol production at ecosystem

scale. Following this concept, momentary net ecosystem exchange (NEE) and number concentration of negative intermediate ions at 2.0-2.3 nm ($N_{neg}$) were analysed for boreal and hemi-boreal ecosystems across Finland and in Estonia. $N_{neg}$ can tell us how effectively biogenic emissions from an ecosystem initiate the new particle formation. Four forests, three agricultural fields, an open peatland, an urban garden, and a coastal site were included focusing on summertime.

We compared the NEE and $N_{neg}$ at each site to the Hyytiälä forest as it is the dominant ecosystem type in Finland. $N_{neg}$ was highest at the urban garden and lowest at the coastal site. The agricultural fields had higher or similar net $CO_2$ uptake rate and higher $N_{neg}$ than all studied forests. The median net $CO_2$ uptake rate of the open peatland was only 31% of that in Hyytiälä, while the median $N_{neg}$ was 77% of that in Hyytiälä. The median net $CO_2$ uptake rate in the urban garden was 63% of that

in Hyytiälä, implying the importance of urban green areas in $CO_2$ sequestration. The coastal site was a minor $CO_2$ source. Considering the combined effect of $CO_2$ uptake and aerosol formation and the large area of forests in Finland, the forests are the most important ecosystems helping to mitigate climate warming.

## 1. Introduction

Carbon dioxide ($CO_2$) is one of the most abundant greenhouse gases in the atmosphere and the most important cause of global warming (e.g. Jia et al., 2022). Terrestrial ecosystems have an essential role in the global $CO_2$ budget through carbon uptake from the atmosphere by photosynthesis and its consequent sequestration to various pools (Walker et al., 2021; Friedlingstein et al., 2022). Globally, the net terrestrial ecosystem uptake of $CO_2$ (i.e. the net

carbon sink) is 3.1 Gt C yr$^{-1}$, which accounts for 32% of $CO_2$ emissions from fossil fuel combustion (Friedlingstein et al., 2022). Terrestrial carbon sequestration, i.e., the process of storing carbon in a carbon pool (IPCC 2022), takes place in both belowground carbon storages (Walker et al., 2021; and the reference therein). Belowground storage includes soil carbon pools, while aboveground storage is primarily in biomass. As a transition between land and open ocean, the coastal

environment is identified as an import carbon sink and estimated to uptake 0.4 Gt C yr$^{-1}$ (Regnier



et al., 2022). Large spatiotemporal variation of continental $CO_2$ uptake is assumed due to different ecosystem and land-use types, climatic conditions, and management pathways (Chang et al., 2021; Friedlingstein et al., 2022). The challenge of increasing the carbon sequestration of ecosystems has been attracting more and more attention with the global goal of reducing $CO_2$ concentrations

in the atmosphere.

Apart from acting as $CO_2$ sinks, terrestrial ecosystems can influence climate by contributing to the formation of new aerosol particles (Kulmala et al., 2004; Kulmala et al., 2014; Kulmala et al., 2020; Yli-Juuti et al., 2021; Junninen et al., 2022; Petäjä et al., 2022, Räty et al., 2023). Globally, aerosols have been reported to induce a net climate cooling effect. The bets estimate of the

effective radiative forcing is $-1.06$ W m$^{-2}$ (Jia et al., 2022). However, large uncertainties exist in the aerosol net radiative forcing estimation, which is tightly associated with the large spatiotemporal heterogeneity in their origin, number concentration and chemical properties.

Atmospheric new particle formation (NPF) is an important source of cloud condensation nuclei (CCN) (e.g. Gordon et al., 2017; Ren et al., 2021; Zhang et al., 2023), which significantly

contributes to aerosol-cloud and aerosol-radiation interaction (Rosenfeld et al., 2014; Ezhova et al., 2018, Artaxo et al., 2022; Petäjä et al., 2022). NPF takes place frequently in many environments, such as forests, urban cities, and coastal areas (e.g. Kerminen et al., 2018; Nieminen et al., 2018; Zheng et al., 2021). It has been reported that NPF is greatly enhanced due to the emission of biogenic volatile organic compounds (BVOCs) in boreal forests and peatlands (Junninen et al.,

2022; Petäjä et al., 2022). Notably, NPF events often take place regionally, extending over distances up to over 1000 kilometres (Kerminen et al., 2018). Multiple types of ecosystems may contribute to the NPF events in a region depending, for example, on the diversity of land use types. It remains unclear whether and how various ecosystems differ in their contributions to regional NPF, and what is the magnitude of such differences.

To overcome the challenge of analysing the role of local ecosystems in regional aerosol formation, the concept of 'CarbonSink+ potential' was recently established (Kulmala et al., 2024). The CarbonSink+ potential enables a direct, ecosystem-scale comparison of $CO_2$ uptake and the intensity of local intermediate ion formation (LIIF) in the atmosphere at the ecosystem scale. The LIIF can be approximated as the number concentration of negative intermediate ions in 2.0-2.3 nm

size range (Tuovinen et al., 2024), to which the aerosol formation at 3-6 nm size range is proportional (Kulmala et al., 2024). The survival probability of small aerosol particles, which





describes the probability of a single particle growing to a certain size without being scavenged, is generally high for particles from 6 nm to CCN size in rural and remote environments (Kulmala et al., 2024; Stolzenburg et al., 2023). The local contributions of certain ecosystems to regional aerosol formation can thus be quantified by LIIF.

This study utilized long-term datasets of intermediate ion concentrations and $CO_2$ fluxes from various boreal and hemi-boreal ecosystems across Finland and in Estonia. In summary, four forests, one open peatland, three agricultural fields, one urban garden, and one coastal site were investigated. The negative intermediate ion concentrations and $CO_2$ fluxes for these ecosystems were compared in different seasons with a focus on the summer. Based on the CarbonSink+ potential concept (Kulmala et al., 2024), the potential of these ecosystems to mitigate climate warming regarding $CO_2$ uptake and aerosol production is discussed.

## 2. Method

### 2.1 Site description

In this study, various ecosystem types, including forests, open peatland, agricultural fields, coastal area, and an urban garden were studied (Figure 1; Table 1). All stations are utilizing the SMEAR (Station for Measuring Ecosystem-Atmosphere Relations; Hari and Kulmala, 2005) concept. The detailed location, ecosystem type, meteorological characteristics and soil type for each site are presented in Table 1. The SMEAR I in Värriö in northern Finland and SMEAR II in Hyytiälä in southern Finland are forest sites both dominated by Scots pine (Kulmala et al., 2019; Kolari et al., 2022), while the forests in Ränskälänkorpi and SMEAR Estonia at Järvselja are mixtures of coniferous and broadleaf trees (Table 1). While Hyytiälä and Värriö are upland forests, i.e., growing on mineral soil, Ränskälänkorpi is a drained-peatland forest (Laurila et al., 2021) and Järvselja has a mosaic of drained swamp, drained peat, and leached gleyic pseudo-podzols (Kangur et.al., 2021; Noe et al., 2015). Two of the agricultural (SMEAR-Agri) sites, i.e. Haltiala, a cereal cropland and Viikki, a managed grassland which was renewed in 2023 with a cereal crop (Pihlatie et al., in preparation), are located in Helsinki. The third agricultural site, Qvidja, is a managed grassland located in southwest Finland (Heimsch et al., 2021). The SMEAR II site at Siikaneva is an open, pristine peatland site ~5 km southwest from the Hyytiälä forest site (Rinne et al., 2018). The SMEAR III at Kumpula, Helsinki is an urban background site and the University of Helsinki





botanical garden, and the city of Helsinki allotment garden are located in the southwest of SMEAR III station with high fraction of vegetation (Järvi et al., 2012). The coastal site is in Tvärminne Zoological Station (TZS). TZS is a 600-ha nature reserve at the Gulf of Finland entrance (northern Baltic Sea), southwest Finland (Virtasalo et al., 2023; Vähä et al., 2024). During the measurement

period, the annual mean temperature for these sites ranged between 0.4 and 7.2°C, while the annual precipitation ranged between 500 and 750 mm (Table 1). SMEAR Estonia, Tvärminne, and Qvidja belong to hemi-boreal climate, while the other ecosystems are characterized by boreal climate.

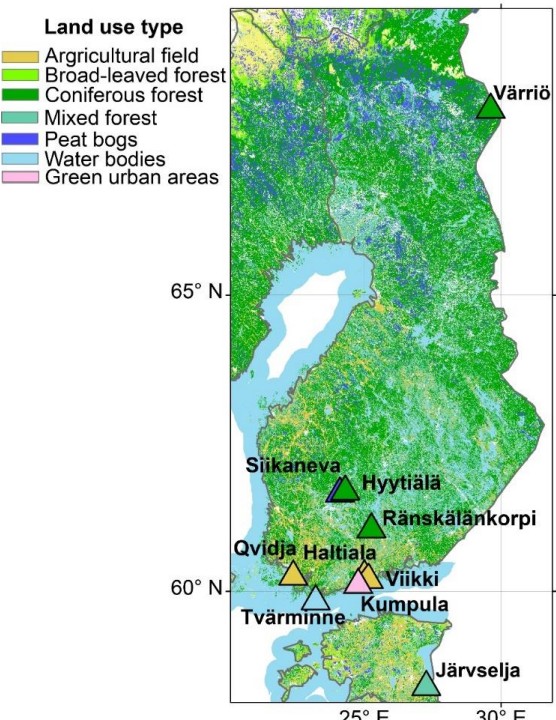

Figure 1. Land type distribution across Finland (Copernicus Land Monitoring Service 2018) and the studied sites with their ecosystem type shown.




Table 1. Meteorological and other main characteristics of the studied sites.

| Stations | | Location | Selected period | Mean air temperature (°C) | Rainfall (mm/yr) | Dominant plant species | Peak LAI | Soil type |
|---|---|---|---|---|---|---|---|---|
| Forest | Hyytiälä, SMEAR II | 61°51'N, 24°17'E | 11/2009-12/2022 | 4.8 | 709[1] | Scots pine and Norway spruce | 4.6 | Haplic podzol |
| | Värriö, SMEAR I | 67°46'N, 29°35' E | 3/2019-12/2022 | 0.4 | 601[2] | Scots pine | 3.2 | Haplic podzol |
| | Ränskälän-korpi | 61°10'N, 25°16'E | 4/2021-12/2022 | 5.4 | 600[3] | Norway spruce, Scots pine, downy birch | ---- | Drained peat |
| | Järvselja | 58°16'N, 27°16'E | 10/2016-12/2020 | 6.8 | 500-750[4] | Birch species, Scots pine, Norway spruce | 6 | Pseudo podzolic |
| Agricultural fields | Haltiala, SMEAR Agri | 60°16'N, 24°57'E | 6/2021-12/2022 | 6.5 | 700[5] | Oat | 5.5 | Silty clay |
| | Qvidja | 60°18'N, 22°24'E | 12/2018-8/2022 | 7.0 | 679[6] | Timothy, meadow fescue | 6.2 | Clay loam |
| | Viikki, SMEAR Agri | 60°13'N, 25°01'E | 7/2022-6/2023 | 6.5 | 792[5] | Timothy (2022), Barley (2023) | 5.2 | Clay loam |
| Peatland | Siikaneva, SMEAR II | 61°50'N, 24°12'E | 11/2019-12/2022 | 5.0 | 710[7] | Moss and sedges | 0.6 | Peat |
| Urban garden | Kumpula, SMEAR III | 60°12'N, 24°58'E | 5/2016-12/2022 | 6.3[5] | 731[5] | Mixed | ------ | ------- |
| Coastal area | Tvärminne | 59°51'N, 23°15'E | 6/2022-8/2023 | 7.2[5] | 639[5] | Seagrass and seaweed | ----- | Sediments |

[1] Neefjes et al. (2022); [2] Kulmala et al. (2019); [3] Laurila et al. (2021); [4] Noe et al. (2015); [5] Finnish Meterology Insititute, only data at the same calendar year of selected period and same or nearby stations as NAIS and eddy covariance measurements were applied; [6] Heimsch et al. (2021); [7] Rinne et al. (2018); ---- data not available.



## 2.2 Atmospheric measurements: intermediate ions, $CO_2$ flux, and meteorological parameters

The number concentration of ions and particles and net ecosystem exchange of $CO_2$ (NEE) were

measured using a Neutral cluster and air ion spectrometer (NAIS, Airel Ltd; Mirme and Mirme, 2013) and eddy covariance method (Aubinet et al., 1999), respectively. The meteorological data, e.g., air temperature, air humidity, and photosynthetic photon flux density (PPFD), were measured simultaneously at same heights with the eddy covariance setup. If the meteorological measurement at the same height was not available, it was replaced by the one from the nearest height. The types

of analysers and detectors at each site are listed in Table S1.

The NAIS is capable of continuous monitoring of ion and total particle concentrations and size distributions over the diameter range of 0.8-42 nm. The ions can be divided into three different size ranges, namely small ions (also named as cluster ions) in sub-2 nm size range, intermediate ions (2-7 nm), and large ions (>7 nm; Tammet et al., 2014). The time resolution was set to five

minutes to optimize signal-to-noise ratio (Mirme and Mirme, 2013). The data were cleaned and quality-checked, considering e.g. the potential interference of rainfall and snow events on the measurements (Manninen et al., 2016). The ion and total particle concentration were further averaged over half an hour.

In this study, we identified the concentration of negative intermediate ions, specifically within the

range of 2.0-2.3 nm ($N_{neg}$), as an indicator of the local intermediate ion formation (LIIF). It is important to note that the intensity of LIIF can serve as an estimate of the local contribution to the regional NPF (Kulmala et al., 2024). It has been observed that $N_{neg}$ displays distinct difference between new particle formation and non-formation periods of intermediate ions (2-7 nm; Tuovinen et al., 2024), thereby making $N_{neg}$ a reliable indicator of LIIF. Moreover, the measurement of

negative intermediate ions between 2.0 and 2.3 nm by NAIS provides a relatively high degree of accuracy, and their footprints are constrained within the ecosystem scale (sub-1 km; Tuovinen et al., 2024; Kulmala et al., 2024). Moreover, the median values of $N_{neg}$ between 00:00 and 06:00, i.e. outside the active hours of the ecosystem, were taken as the background concentration at each site. The background value of $N_{neg}$ was calculated separately for each season. A narrower time

window for background concentration compared to the one proposed by Aliaga et al. (2023), 21:00-06:00, was applied due to the more northern site Värriö with longer day length in the summer in this study. We then calculated the changes of $N_{neg}$ ($\Delta N_{neg}$) by subtracting the background concentration in each season from $N_{neg}$. The diurnal variation of median $\Delta N_{neg}$ were



presented together with $N_{neg}$ (Section 3). The use of $\Delta N_{neg}$ was assumed to eliminate the influence

of background clustering at different sites, so that it reflects the intensity of negative intermediate

ion production from the specific ecosystem.

The eddy covariance measurement of $CO_2$ fluxes is based on the turbulence theory, i.e. assumption

that the turbulent flux remains relatively stable in a constant flux layer above the canopy (Lee and

Hu, 2002), and it is equal to the covariance of vertical wind speed and ambient $CO_2$ concentration

in flat and horizontally homogeneous surface (Aubinet et al., 1999). The measurement system

requires a fast-response analyser of the $CO_2$ concentration (10 Hz) and 3-D sonic anemometer.

The raw eddy covariance 10 Hz-data were pre-processed with standard steps, including despiking,

detrending, dilution correction and 2-D coordinate rotation (Aubinet et al., 1999). The fluxes were

further lag-time adjusted and corrected for spectral loss (Aubinet et al., 1999). Either EddyUH

(Mammarella et al., 2016) or EddyPro (Fratini and Mauder, 2014), or the program introduced by

Heimsch et al. (2021) were applied for the pre-processing for one site. The processed fluxes were

accepted only if they met the stationarity and developed turbulence criterion (Foken and Wichura,

1996) exceeding the site-specific friction velocity thresholds (Table S1). The quality-checked $CO_2$

fluxes at the forest sites were further partitioned into gross primary production (GPP) and

175 ecosystem respiration ($R$) using site-specific dependence of $R$ on the air and/or soil temperature

and GPP on the PPFD and air and/or soil temperature (Kulmala et al., 2019).

**2.3 Data selection criteria**

In this study, the analyses were restricted to periods when both negative intermediate ion

concentration and NEE were available (Table 1). Therefore, different time periods were applied

for each of different sites. For Hyytiälä, Värriö, Järvselja, Qvidja, Siikaneva, and Kumpula sites,

the long-term data were available for more than 3 years. At Hyytiälä, 12 years of continuous

observations were used. For the sites with recently established atmospheric measurement,

Tvärminne, Ränskälänkorpi, Haltiala and Viikki - data were available for approximately one to

one and a half years. In total, 35 site-years of data were utilized in this study. As we focused on

the potential of the ecosystem to uptake $CO_2$ and form intermediate ions, the inter-annual variation

at the sites was not discussed in this study (Kulmala et al., 2019; Alekseychik et al., 2021; Heimsch

et al., 2021).

 

Due to the thinning of Hyytiälä forest in the beginning of year 2020, when 30% of tree basal area was removed (Aalto et al., 2023), data from that year were discarded from the analyses to exclude
the immediate thinning effect on the studied variables. In the Ränskälänkorpi forest, the western part of the site was selectively harvested (~60% of basal area removed) and the eastern part of the site was clear-cut in the spring and summer of 2021 with control site left in the middle. The NAIS equipment was located in the border between the control and clear-out. The location was ~230 m east from the eddy covariance tower located in the border between control and selective harvested
sites. In this study, only data with wind blowing from the selective harvested site from the west (WD>180°) and wind speed higher than 2 m s$^{-1}$, were considered. Note that carbon removed from the site in harvested tree biomass is not accounted in the measured flux of $CO_2$. At Kumpula site, data from the garden area, i.e., 180°-320°, were utilized (Järvi et al., 2012).

At the agricultural sites, the management activity is relatively intense and can distinctly influence
the $CO_2$ fluxes (Heimsch et al., 2021). Note that the carbon removed in harvested crop biomass and the carbon added to the site in fertilizers do not directly contribute to the measured net flux of $CO_2$. For the Qvidja site, only measurements from wind direction between 0° and 30° or 140° and 360° were included to avoid interference from the nearby experimental areas. Similarly, at the Viikki site, only measurements from wind direction between 145° and 245° were included in the
analysis to avoid data from other nearby fields.

The open peatland at Siikaneva is surrounded by forests. By applying a footprint model (Kljun et al., 2015), 90% of the $CO_2$ flux footprint is within ~200 m from the measurement tower, i.e., constrained within the peatland. At the coastal Tvärminne site, the NAIS instrument trailer is on the shore, and the eddy covariance mast is on an island, ~110 m east of the shore. Only data with
210 wind direction from 95° to 165° and from 205° to 240°, i.e., from the coastal water without being disturbed by trees on nearby islands, were included in the analysis at this site.

## 3. Results and discussion

### 3.1 Comparison of momentary NEE in different ecosystems

The diurnal variation of NEE between the studied forests, urban garden area, agricultural fields,
open peatland, and coastal site in spring (MAM) and summer (JJA) are presented in Figures 2-4.



The corresponding comparison in the autumn (SON) and winter (DJF) are presented in Figures S1-S3.

For the forest sites, the hemi-boreal Järvselja site tended to have the highest net $CO_2$ uptake rate (absolute values of NEE when it is negative) at midday (10:00-14:00) in both spring and summer.

The median net $CO_2$ uptake rate at midday in Järvselja forest reached 12.2 μmol $m^{-2}$ $s^{-1}$ in summer. The lowest net $CO_2$ uptake rate at midday was found in the northern Värriö site, with the median being 4.69 μmol $m^{-2}$ $s^{-1}$. This difference may be due to the higher air temperature in the hemi-boreal Estonian site and lower temperature at Värriö (Figure S4), as the ecosystem productivity at high latitudes in Europe is typically temperature limited (Yi et al., 2010).

In summer, the net $CO_2$ uptake rate in the urban garden area at Kumpula was comparable with the drained peatland forest in Ränskälänkorpi. In the other seasons, the urban garden area was a net source of $CO_2$ most of the time, similar to the results previously reported for the years 2006-2010 from the same site (Järvi et al., 2012).



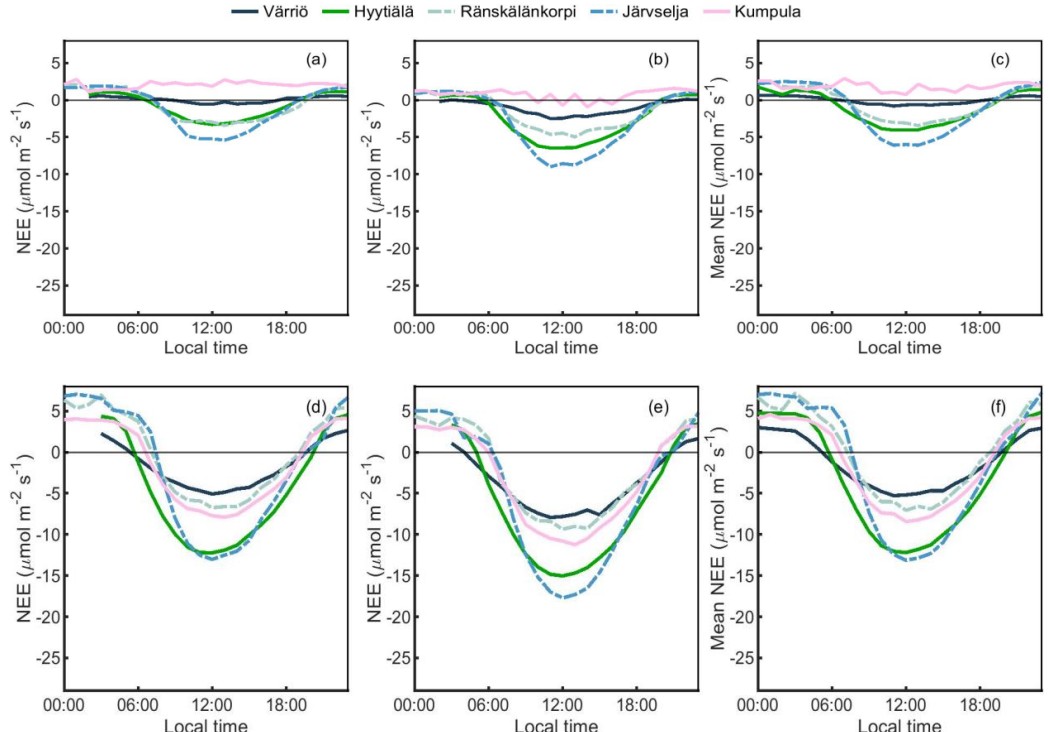

Figure 2. The $50^{th}$ percentile (a), $25^{th}$ percentile (b), and mean values (c) of NEE at each hour for the forest sites and urban garden in spring (MAM) and the corresponding $50^{th}$ percentile, $25^{th}$ percentile, and mean values in summer (JJA), (d), (e), (f), respectively.

In the case of agricultural fields in summer (Figure 3), the Haltiala site had higher momentary net $CO_2$ uptake than the other two agricultural sites. Notably, in spring, the croplands in Viikki and Haltiala were net sources of $CO_2$, while the grassland in Qvidja was a $CO_2$ sink during daytime

with a similar uptake rate to the Hyytiälä forest. The different plant species (Table 1) and management activities between the agricultural fields likely caused the differences in their seasonal $CO_2$ fluxes. The upper quartile of the momentary net $CO_2$ uptake, i.e., absolute values of $25^{th}$ percentile NEE, was also about two times higher in Haltiala cropland than in Hyytiälä forest in summer. The midday momentary net $CO_2$ uptake rate in Viikki cropland was slightly higher

than that in Hyytiälä forest, while that in Qvidja agricultural grassland was slightly lower than in Hyytiälä. It is also important to note that the harvests of plant biomass decreased local carbon



storage which was not accounted for in the measured $CO_2$ fluxes. Qvidja and Viikki agricultural sites were harvested twice in summer and the harvest in Haltiala cropland was done only at the end of the growing season, whereas the typical rotation length in managed boreal are 60-100 years

in Southern Finland.

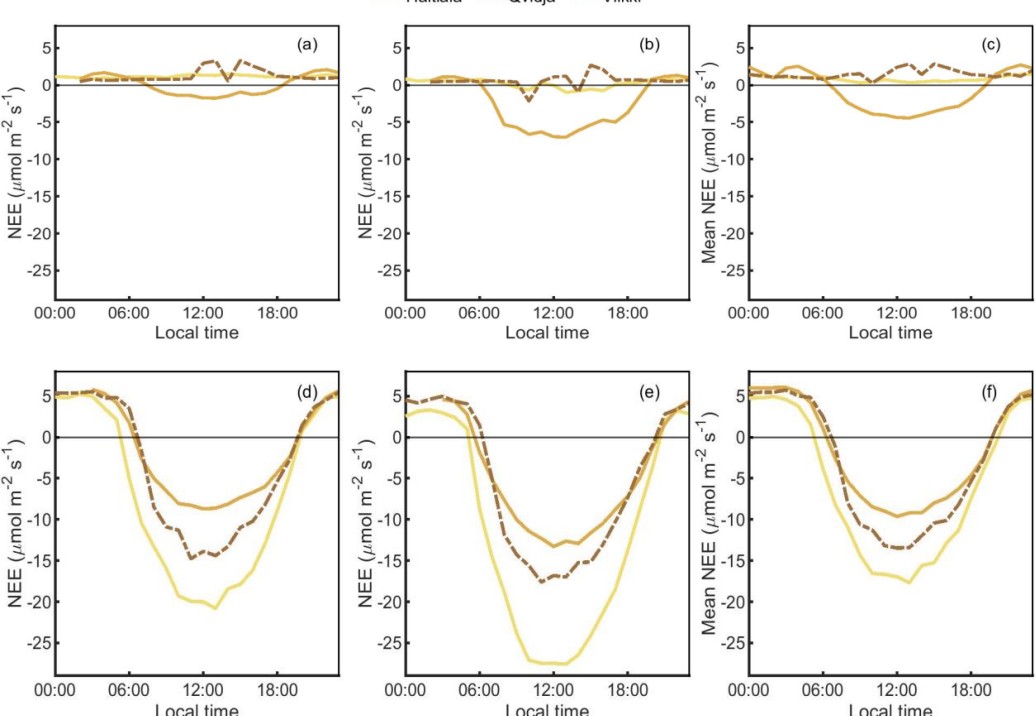

Figure 3. The $50^{th}$ percentile (a), $25^{th}$ percentile (b), and mean values (c) of NEE at each hour for
the agricultural fields in spring (MAM) and the corresponding $50^{th}$ percentile, $25^{th}$ percentile, and mean values, (d), (e), (f), in summer (JJA), respectively.

The $CO_2$ uptake rate and respiration rate (nighttime $CO_2$ fluxes) in the open peatland and coastal area (Figure 4) were much lower than those in the agricultural fields and forests during spring and
summer. Still, the Siikaneva peatland remained a weak net sink of $CO_2$ during daytimes in all the seasons except in winter. The midday NEE at Tvärminne were -0.26 and 0.01 $\mu mol\ m^{-2}\ s^{-1}$ in spring and summer, respectively. Hence, net $CO_2$ uptake possibly appears in spring in this Baltic coastal area under certain conditions, i.e., when the partial pressure of $CO_2$ in the water is lower



than that in the air (Roth et al., 2023). This may be induced by phytoplankton and submerged

vegetation $CO_2$ uptake in the spring (Roth et al., 2023).

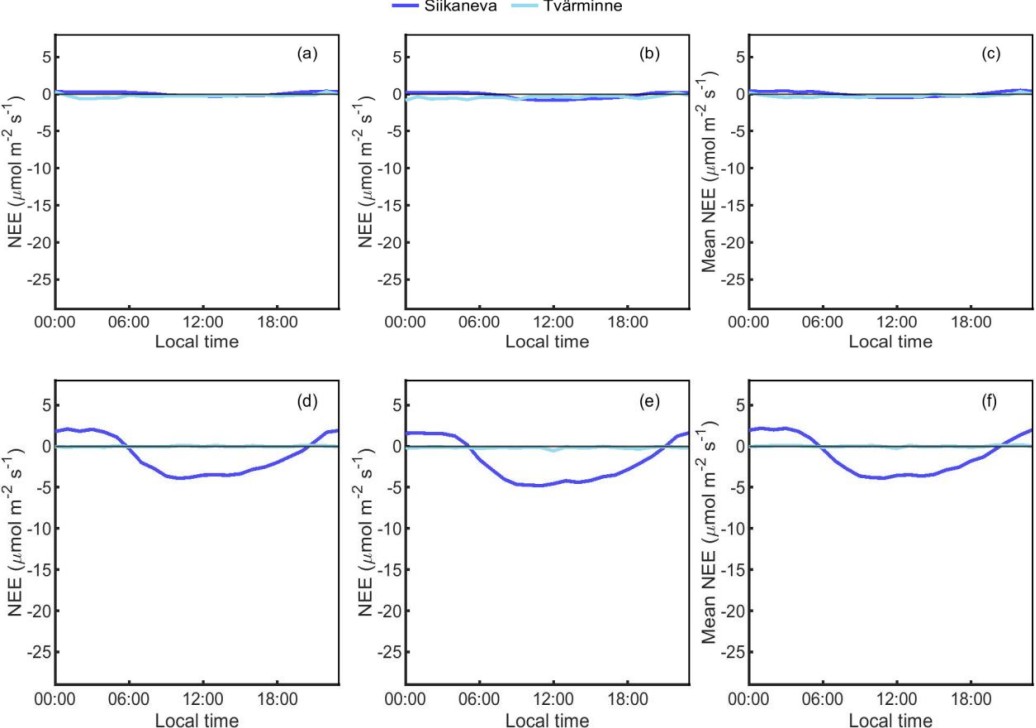

Figure 4. The 50[th] percentile(a), 25[th] percentile (b), and mean values (c) of NEE at each hour for

the peatland and coastal area in spring (MAM) and the corresponding 50[th] percentile, 25[th] percentile, and mean values in summer (JJA), respectively.

Additionally, the Ränskälänkorpi and Järvselja forests turned into a $CO_2$ source 1-2 hours earlier in the late afternoon of summer than the other two forests (Figure 2). Note that the soil at

Ränskälänkorpi and Järvselja is mainly drained peatland and water-logged soil (Table 1), respectively, which is indicated by high organic carbon content (Laurila et al., 2021; Noe et al., 2015). The higher air temperature and soil organic carbon content may drive higher respiration at the two sites, which is reflected in the nighttime fluxes (Figure 2). Hence, even though the GPP at Järvselja and Ränskälänkorpi in the late afternoon were close to that at Hyytiälä forest (Figure 5),




net emissions of $CO_2$, i.e., positive NEE values, were observed at these two forest sites in the
earlier and later hours of the day.

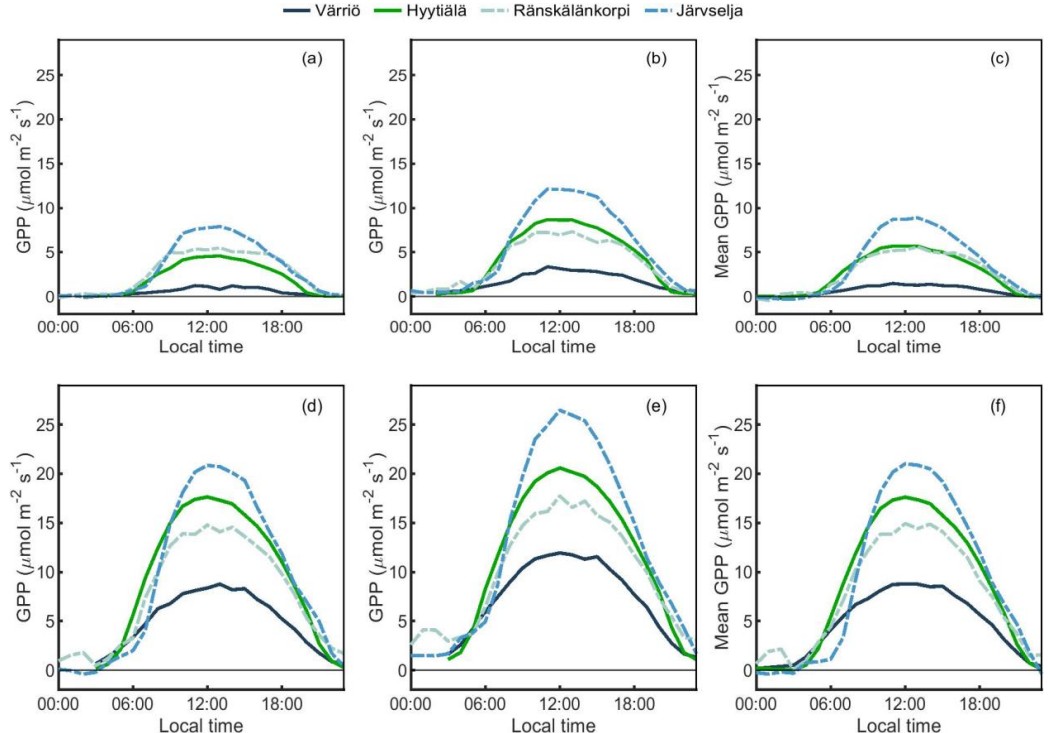

Figure 5. The 50th percentile (a), 75th percentiles (b), and mean values (c) of GPP at each hour for
the forest sites in spring (MAM) and the corresponding 50th percentile, 75th percentile, and mean
values in summer (JJA), (d), (e), (f), respectively.

## 3.2 Comparison of negative intermediate ion concentrations across different ecosystems

The comparison of $N_{neg}$ between different ecosystems in spring and summer are presented in
Figures 6-8. It was assumed that negative intermediate ions at 2.0-2.3 nm can describe how
efficiently the ecosystem can produce new aerosol particles (Kulmala et al., 2024; Tuovinen et al.,
2024). The corresponding values of $N_{neg}$ in autumn and winter were much lower than those in
spring and summer (Figures S5-S7). The median values of $N_{neg}$ in the daytime in spring were

higher than those in the Haltiala and Viikki croplands, Siikaneva peatland, and Kumpula urban





garden area. At the other sites, summer median values were higher. In contrast, the difference between $75^{th}$ and $50^{th}$ percentile of $N_{neg}$ in spring was higher than that in summer in all the studied sites. The larger upper quartile deviation of $N_{neg}$ in spring implied that the LFII were either more frequent or stronger in spring than in summer at all the sites (Dal Maso et al., 2005; Dada et al.,

2018; Nieminen et al., 2018). For all the sites, the diurnal variation of negative intermediate ions in spring and summer was clear, i.e., a distinct peak in the daytime. In the winter, the diurnal cycle of $N_{neg}$ was not visible in any of the studied sites (Figures S6-S8). This agrees with the observation that the global radiation and air temperature are observed to correlate positively with concentration of negative intermediate ions at 2-4 nm in the Hyytiälä boreal forest (Neefjes et al., 2022).

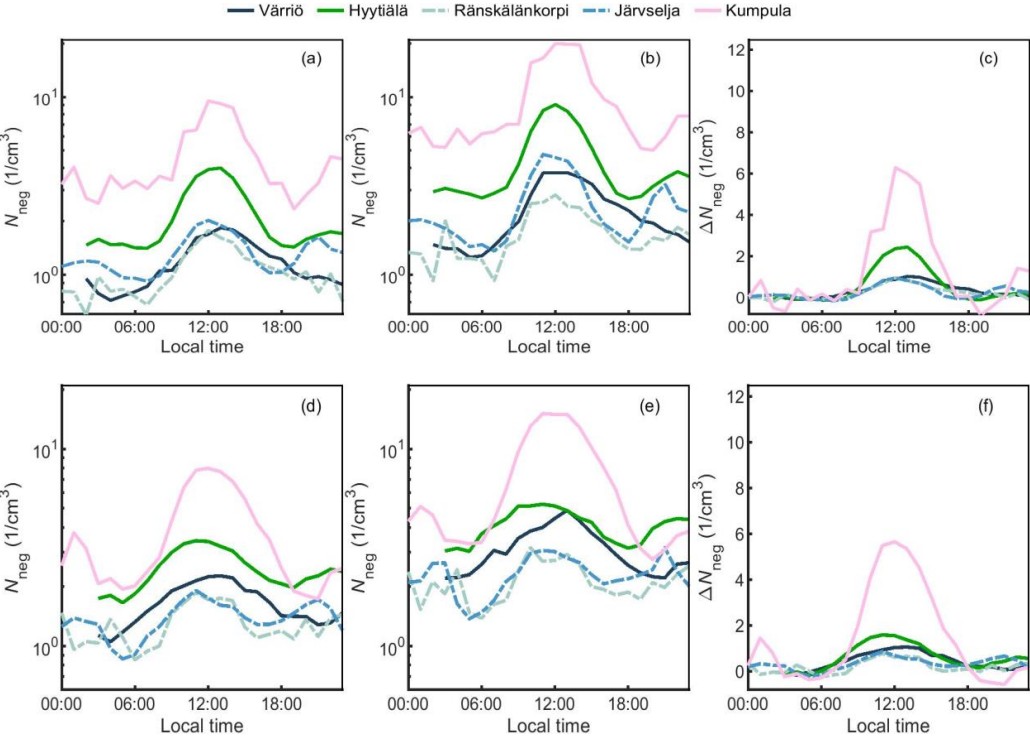

Figure 6. The $50^{th}$ percentile (a) and $75^{th}$ percentile (b) of negative intermediate ions ($N_{neg}$) at 2.0-

2.3 nm ($N_{neg}$) at each hour and the daily fluctuations of $N_{neg}$ (c) for the forests and urban garden in spring (MAM) and the corresponding $50^{th}$ percentile, $75^{th}$ percentile, and normalized concentration for median values in summer (JJA), (d), (e), (f), respectively.



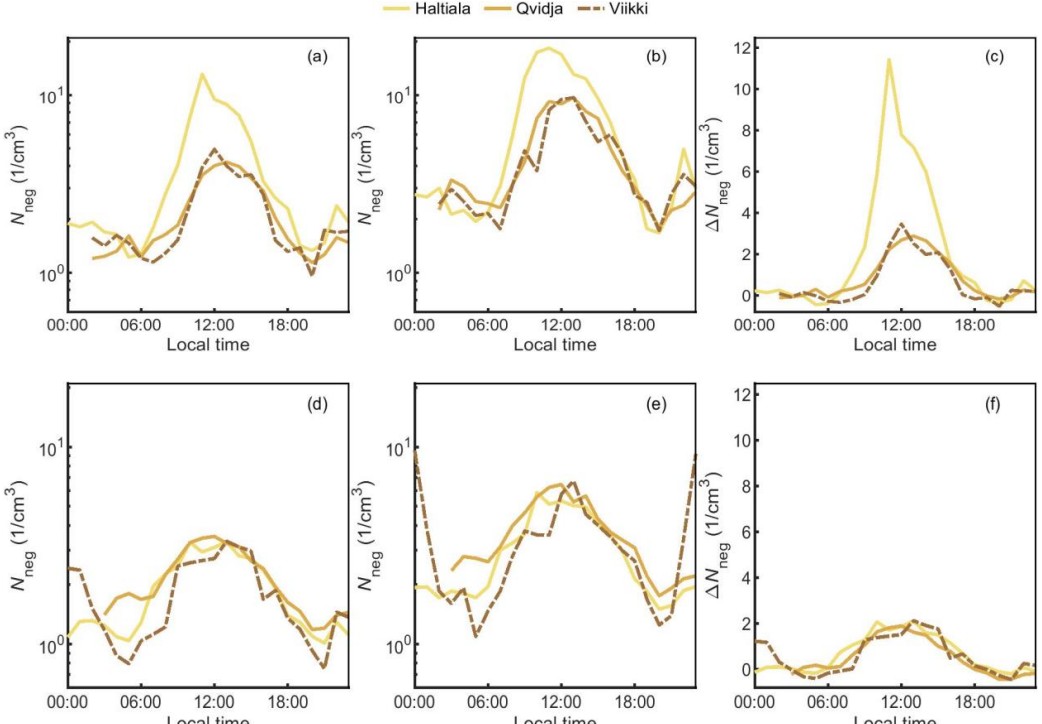

Figure 7. The 50th and 75th percentile (b) of negative intermediate ions ($N_{neg}$) at 2.0-2.3 nm at each hour and the daily fluctuations of $N_{neg}$ (c) for the agricultural fields in spring (MAM) and the corresponding 50th percentile, 75th percentile and normalized concentration for median values, (d), (e), (f), in summer (JJA), respectively.



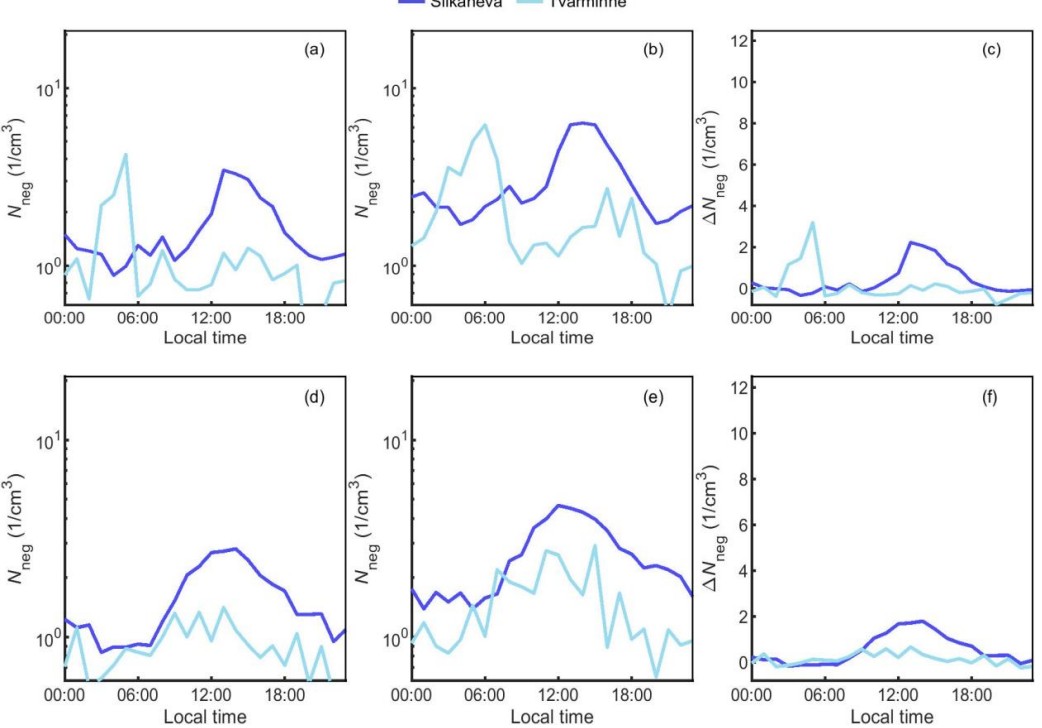

Figure 8. The 50th percentile (a) and 75th percentile (b) of negative intermediate ions ($N_{neg}$) at 2.0-
2.3 nm at each hour and the daily fluctuations of $N_{neg}$ (c) for the peatland and coastal area in spring
(MAM) and the corresponding 50th percentile, 75th percentile, and normalized concentration for
median values in summer (JJA), (d), (e), (f), respectively.

The daily fluctuations of $N_{neg}$ ($\Delta N_{neg}$) were calculated by subtracting the background concentration
from $N_{neg}$ in each season (Section 2.2). In spring, median $\Delta N_{neg}$ in the midday for the forests ranged
between 0.8 and 2.0 cm$^{-3}$ (Table S2), with the lowest value in Järvselja and the highest value in
Hyytiälä forest. The midday mean $\Delta N_{neg}$ at the Kumpula urban garden area was 4.9 cm$^{-3}$, which
was higher than in any of the studied forests. The presence of more abundant nucleation precursors
at the Kumpula urban area may facilitate the ion formation (Nieminen et al., 2018). In summer,
$\Delta N_{neg}$ decreased compared to spring at all the sites except Siikaneva peatland and Tvärminne
coastal areas. Seasonal changes in the clustering precursors and their dependence on air
temperature and radiation may drive the seasonal variation of $\Delta N_{neg}$ at all the sites.





It is notable that all the agricultural sites had higher midday $\Delta N_{neg}$ than the forest sites in spring, varying between 2.3 and 7.7 cm$^{-3}$. The application of fertilizers in agricultural fields is known to remarkably increase the atmospheric concentration of ammonia (NH$_3$) (Olin et al., 2022). NH$_3$ can

stabilize the critical clusters in the nucleation process driven by sulfuric acid (H$_2$SO$_4$) (Kulmala et al., 2013). H$_2$SO$_4$ in the air is majorly formed by oxidation of sulphur dioxide, which can be transported from a longer range than the intermediate ions. However, the frequency of NPF events was found not to increase after the fertilization in Qvidja grasslands (Dada et al., 2023). Similarly, the frequency of daytime NPF events did not correlate with agriculture activities in a cropland in

France (Kammer et al., 2023). Dada et al. (2023) observed that NH$_3$, H$_2$SO$_4$, and low volatile organic compounds originating from BVOC oxidation play a synergistic role in clustering in Qvidja, resulting in a higher formation rate and number concentration of particles than in Hyytiälä forest. Note that since the Haltiala and Viikki croplands are located in Helsinki, the nucleation precursors and thereby the nucleation rate may be enhanced by anthropogenic pollution in the city.

The exact reasons why there were higher $N_{neg}$ and $\Delta N_{neg}$ at these agricultural sites require more measurement of the clustering precursors.

Furthermore, in spring and summer, the night-time $N_{neg}$ increased again at around 20:00 for all the sites, suggesting a ubiquitous nighttime clustering in warm seasons (Mazon et al., 2016). Moreover, in summer, the 75$^{th}$ percentile of nighttime $N_{neg}$ at Viikki was comparable with the daytime $N_{neg}$.

The decreased boundary layer height (Chen et al., 2016; Neefjes et al., 2022), especially in clear nights, may also facilitate the accumulation of formed clusters and eventually lead to the nighttime peak.

### 3.3 Potential of different ecosystems to contribute to CO$_2$ uptake and negative intermediate ion production

Since we aimed to compare the potential of ecosystems for net CO$_2$ uptake and local production of negative intermediate ions (LIIF), the most active periods for the ecosystem plants are discussed in detail in this section, i.e., midday in summertime. The potential of the studied ecosystems for net CO$_2$ uptake and LIIF at midday during summertime are listed in Table 2. For median values in summer, $N_{neg}$ was found to be highest in the urban garden, followed by the agricultural fields

(Figure 9). The agricultural fields generally had higher $N_{neg}$ than the studied forests. The open peatland had lower $N_{neg}$ than Hyytiälä forest but higher than the other forests. The $N_{neg}$ at the



coastal area was the lowest. The momentary net $CO_2$ uptake rate at midday in summer was highest in agricultural fields, followed by the forests. The urban garden in this study displayed distinct net $CO_2$ uptake, lower than the forests and higher than the open peatland. The coastal area at midday in summer was a very weak $CO_2$ sink. In the urban garden area in Kumpula, median $N_{neg}$ was 2.2 times of that in Hyytiälä forest, while the median NEE only reached 63% of that in Hyytiälä forest. The variation of momentary NEE and $N_{neg}$ were distinct even between a similar type of ecosystem in a similar latitude, e.g., within forests and agricultural fields. For forests, the most southern Järvselja had the highest net $CO_2$ uptake rate, while the median $N_{neg}$ in the midday in summer was similar to Ränskälänkorpi and 53% of that in Hyytiälä forest. Hyytiälä forest had higher $N_{neg}$ than the other forests. For agricultural sites, the net $CO_2$ uptake rate at Qvidja and Viikki were close to that in Hyytiälä forest, while it was much higher in Haltiala croplands than in Hyytiälä forest. On the contrary, the $N_{neg}$ were highest in Qvidja between the three agricultural sites, and median $N_{neg}$ in the other two sites were slightly smaller than in Hyytiälä forest.

Another potent greenhouse gas, methane ($CH_4$) can be emitted through microbial activities in anoxic conditions, e.g., peatlands and coastal areas (Mathijssen et al., 2022; Roth et al., 2023). Considering that $CH_4$ has a sustained-flux global warming potential 45 times of $CO_2$ over 100 years (Roth et al., 2023; and the reference therein) , the net $CO_2$ equivalent emission of $CH_4$ is estimated 2.5-8.6 times of $CO_2$ uptake in Siikaneva peatland (Mathijssen et al., 2022). $CH_4$ emissions may largely compensate the $CO_2$ uptake in open and non-ditched peatlands. Similarly, the emission of $CH_4$ from coastal environment around Baltic Sea may offset 28% of the $CO_2$ sink in macroalgae-dominated coastal area (Roth et al., 2023). For ions, the summertime midday median $N_{neg}$ at the peatland in Siikaneva was 77% of that in Hyytiälä forests (Table 2). As the open peatland is surrounded by forest within 1 km, the negative ion at 2.0-2.3 nm may be influenced by nearby forests. Also, the terpene emissions from the peatlands can initiate stronger NPF than in the Hyytiälä boreal forest (Junninen et al., 2022; Huang et al., 2024). However, these events were majorly reported to occur at late evening.

The CarbonSink+ potential, especially $CO_2$ uptake, may largely vary within agricultural fields in Finland. Agricultural fields may be highly productive in local formation of negative intermediate ions, affected by their vegetation and management practises. However, considering the much larger area of forests in Finland than that of agricultural fields (Table 2), boreal forests in Finland





in total are likely the largest contributor of climate cooling when considering the $CO_2$ uptake and local new particle formation.

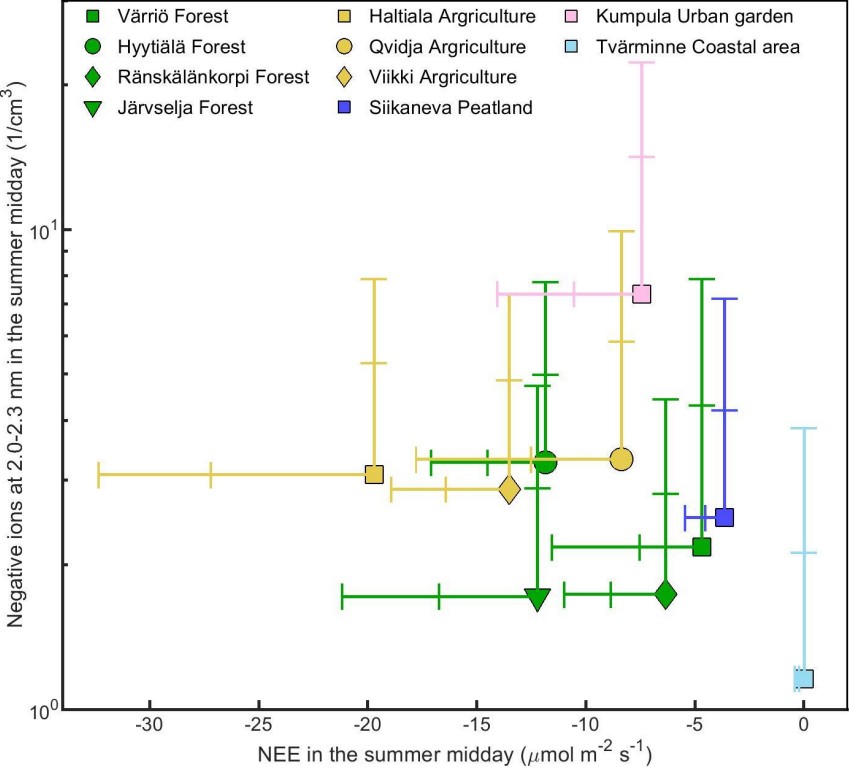

Figure 9. Comparison of NEE and negative intermediate ions at 2.0-2.3 nm at midday in summer between the sites. The error bars for x axis are 10[th] and 25[th] percentile for NEE, while they are 75[th] and 90[th] percentile of the negative intermediate ions at each site for y axis.





Table 2. Comparison of NEE and negative intermediate ions at 2.0-2.3 nm size range across

the hemi-boreal and boreal ecosystems at midday (10:00-14:00) in summer.

| Ecosystem | Site | Area in Finland (ha) | Median $N_{neg}$ (1/cm$^3$) | Median $N_{neg}$/median $N_{neg, Hyytiälä}$ | 75$^{th}$ percentile $N_{neg}$/75$^{th}$ percentile $N_{neg, Hyytiälä}$ | Midday NEE (µmol m$^{-2}$ s$^{-1}$) | Median NEE/ median NEE$_{Hyytiälä}$ | 25$^{th}$ percentile NEE/25$^{th}$ percentile NEE$_{Hyytiälä}$ |
|---|---|---|---|---|---|---|---|---|
| Forest | Hyytiälä | 20.3 million[a] | 3.27 | 1 | 1 | -11.84 | 1 | 1 |
| | Värriö | | 2.18 | 0.67 | 0.87 | -4.69 | 0.4 | 0.52 |
| | Järvselja | | 1.72 | 0.53 | 0.58 | -12.23 | 1.03 | 1.15 |
| Drained peatland forest | Ränskälänkorpi | 4.2 million[a] | 1.74 | 0.53 | 0.57 | -6.35 | 0.54 | 0.61 |
| Agricultural field | Haltiala | 2.3 million[a] | 3.08 | 0.94 | 1.06 | -19.69 | 1.66 | 1.88 |
| | Qvidja | | 3.32 | 1.01 | 1.17 | -8.37 | 0.71 | 0.86 |
| | Viikki | | 2.88 | 0.88 | 0.97 | -13.52 | 1.14 | 1.13 |
| Open peatland | Siikaneva | 0.21 million[c] | 2.51 | 0.77 | 0.85 | -3.65 | 0.31 | 0.31 |
| Urban garden area | Kumpula | ----- | 7.33 | 2.24 | 2.86 | -7.44 | 0.63 | 0.73 |
| Coastal area | Tvärminne | ----- | 1.46 | 0.45 | 0.53 | 0.01 | 0.00 | 0.01 |

[a] Natural Resources Institute Finland 2022; [b] The area of oligotrophic open fens (Turunen and Valpola 2020);

----- data not available

## 4. Conclusions

The CarbonSink+ potential concept was established recently and provides a direct comparison

of local contribution to $CO_2$ uptake and aerosol formation at ecosystem scale. The value of

negative intermediate ion concentration at 2.0-2.3 nm size range ($N_{neg}$) was applied as an

indicator of the corresponding contribution of each ecosystem to produce new aerosol particles

which, after their subsequent growth to larger sizes, are able to cool the atmosphere in a

regional scale. Following this concept, net ecosystem $CO_2$ exchange fluxes (NEE) and $N_{neg}$

were analysed in ten hemi-boreal and boreal ecosystems in Finland and Estonia. The boreal



forest in Hyytiälä was chosen as a reference site, to which the values of NEE and $N_{neg}$ at all

other sites were all compared.

The results showed that the agricultural fields had similar or even higher $CO_2$ uptake potential

compared to Hyytiälä forest during the summer. Note that the decreased carbon storage due to

harvest in the fields was not taken into account in this study. A distinct $CO_2$ uptake in the urban

garden at midday in summer was observed, lower than that in Hyytiälä forest but higher than

observed in the open peatland. The coastal area considered in this study remained a very small

$CO_2$ source during summertime. The differences in $N_{neg}$ between the studied sites were not as

large as those in NEE. Ubiquitous nighttime clustering was observed across the ecosystems.

At midday in summer, $N_{neg}$ was highest in the urban garden, followed by the agricultural fields.

The coastal area had the lowest $N_{neg}$. The forest sites generally had lower $N_{neg}$ than the

agricultural sites. The $N_{neg}$ in the open peatland was lower than Hyytiälä forest but higher than

other studied forests. Note that the urban garden and agricultural sites in Helsinki might be

more influenced by air pollution compared to the forests and open peatland that were

background sites. Overall, considering the large area of forests in Finland and Estonia, the

forests in total have the largest potential of climate cooling when considering the $CO_2$ uptake

and local new particle formation.

**Data availability**

Measurement data at the sites, including ions data, eddy covariance data and meteorological

data, are available upon request from the corresponding author before the relevant databases

are open to the public.

**Author contributions**

ST, JL, and RT were responsible for the ion measurements. PS, AL, MP, AL, MK, HR, LH,

AV, IM, and SN were responsible for the eddy covariance measurement and analysed the raw

data. MK designed the study. PKe, AL, PKo, TN, OP, EE, TK, JB, VMK, and MK analysed

the data and interpreted the results. PKe prepared the firs-draft paper. All authors contributed

to discussion of the results and provided input for the paper.



**Competing interests**

The authors declare no competing interests.

**Acknowledgement**

We acknowledge the following projects: ACCC Flagship funded by the Academy of Finland grant number 337549 (UH) and 337552 (FMI), Academy professorship funded by the Academy of Finland (grant no. 302958), Academy of Finland projects no. 1325656, 311932, 334792, 316114, 325647, 325681, 339489, the Strategic Research Council (SRC) at the Academy of Finland (352431), Jane and Aatos Erkko Foundation, "Gigacity" project funded by Wihuri foundation, European Research Council (ERC) project ATM-GTP (742206), and European Union via Non-CO2 Forcers and their Climate, Weather, Air Quality and Health Impacts (FOCI). This project has received funding from the European Union – NextGenerationEU instrument and is funded by the Research Council of Finland under grant number 347782. University of Helsinki support via ACTRIS-HY is acknowledged. Support of the technical and scientific staff in all sites are acknowledged. For SMEAR Estonia we acknowledge the Estonian Research Council Grant PRG 1674, the Estonian Environmental Investment Centre (KIK) project number 18392 and the European Union's Horizon 2020 Research and Innovation programme (grant agreement no. 871115) ACTRIS IMP. INAR research infrastructure (RI), ICOS RI, ACTRIS RI and eLTER RI are gratefully acknowledged for the continuous ecosystem-atmosphere measurements used in this study.

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
