# Peer review of "Potential of carbon uptake and local aerosol production in boreal and hemi-boreal ecosystems across Finland and in Estonia"

_EGUsphere, 2024_

## Referee Comment (RC2)

Potential of carbon uptake and local aerosol production in boreal and hemi-boreal ecosystems across Finland and in Estonia

Piaopiao Ke et al.,

This manuscript explained and researched the $CO_2$ uptake and local aerosol production at different types of stations in Finland. Authors analysed each NEE and $N_{neg}$ characteristics at each station and compare between them. Introduction is very persuasive and interesting why this study is necessary. However, the way to describe the result of experiments is not kind to the readers. Result was simple and it seems like lack of discussion. I also wonder whether the conclusion gives the answer to the question in the Introduction section. Hopefully authors read the manuscript carefully with reader's view and revise/describe them explicitly.

General comments

1. Title mentioned 'boreal and hemi-boreal ecosystem' but all manuscripts are not linked to boreal and hemi-boreal region. Authors did not explain which site is belonging to boreal or hemi-boreal. According to the category in Table 1, there are forest, agricultural, peatland, urban garden and coastal area. It was hard to match to the title. Authors should consider how to make this manuscript well match to the title and more interesting.

2. Authors did not make station ID and mentioned the station name directly (Table 1). But readers are not familiar with Finish and hard to follow up what type of station with only station's names. Hope authors reconsider the way to explain station' name.

3. In many paragraphs, authors did not suggest table/figure # but also values. For example, *L367-369: The momentary net $CO_2$ uptake rate at midday in summer was highest in agricultural fields, followed by the forests. The urban garden in this study displayed distinct net $CO_2$ uptake, lower than the forests and higher than the open peatland*. Next to the highest, it needs value. Lower than or higher than, authors can suggest certain values. Without certain values, all explanations such as "lower than, higher than, highest, similar" are very boring and insufficient to deliver what authors would like to say.

4. Hope authors can describe the sufficient reason after the result, for example, in section 3.3. This is scientific paper not a report.

Minor comments

1. L58: Räty et al., 2023 was not on the reference list.
2. L86: long-term datasets. What's the definition of 'long-term' here? Only long-term datasets are from Hyytiälä station and others are less than a decade.
3. Table 1: Between the category (such as Forest and Agricultural) it would be good to have a line.
4. Table 1: Hope station name can be reconsider. Especially Hyytiälä station is 'reference station', used everywhere but in the forest category. Author can name stations with their characteristic.
5. Table 1: Location. Hope author can add height of each station.
6. L133: *at same heights*. Please add values.
7. L137: *Diameter range*. Does it mean particle size?
8. L140: *The data were cleaned*. What does it mean? Does it mean 'filtered' or 'selected'?
9. L142: *total particle concentration*. Is it mass concentrations or number concentrations or others?
10. L151: *their footprints are constrained within the ecosystem scale*. Don't authors think that it can be differed according to the height of station?
11. L153: *outside the active hours of the ecosystem, were taken as the background concentration at each site*. This sentence is conflict against L352-357.
12. L194: *tower*. Hope to add the tower height.
13. L203: *interference*. What kind of interference?
14. L205: *fields*. What is the possible source to affect on data?
15. L226: *In the other seasons, the urban garden area was a net source of $CO_2$ most of the time, similar to the results previously reported for the years 2006-2010 from the same site (Järvi et al., 2012)*. Most of section did not explain the reason. Why is that?
16. Figure 2-4: Why did authors add 50[th] and 25[th] graph? It would better to add median with standard deviation to see the variation of the values since experimental periods are different at each station.
17. L237: *In the case of agricultural fields in summer (Figure 3), the Haltiala site had higher momentary net $CO_2$ uptake than the other two agricultural sites*. Why don't authors add specific value for 'higher than' precisely?
18. L239: *Qvidja was a $CO_2$ sink during daytime with a similar uptake rate to the Hyytiälä forest*. Add specific value and figure #.
19. L240: *The different plant species (Table 1) and management activities between the agricultural fields likely caused the differences in their seasonal $CO_2$ fluxes*.
    This is very confusing to readers. In table 1, authors mentioned of plants type but not management activities. I confirmed that it was described in the section 2.1, it would be good to add table 1.

20. L294: *The median values of $N_{neg}$ in the daytime in spring were higher than those in the Haltiala and Viikki croplands, Siikaneva peatland, and Kumpula urban garden area*. I cannot understand what this sentence means.

21. L296: *summer median values were higher*.
    Please finalize the sentence. Higher than what?

22. L338: *The application of fertilizers in agricultural fields is known to remarkably increase the atmospheric concentration of ammonia ($NH_3$)*. This sentence seems to be generalized to all stations. Please add specific type of regions.

23. Figure 9: I cannot understand the meaning of bar in Figure 9.

24. Table 2: There is no explanation of 'c' (superscript) and there is no 'b' in the table.

25. L409: Summary and conclusions or only summary?

26. L416: What are 10 stations belonging to hemi-boreal station? Why did not author define 'hemi-boreal' and 'boreal' in the manuscript?

27. L421-423: Why didn't authors add the reasons?

28. L431: What's the difference between reference station and background station? Please add the definition.

---

## Author Comment (AC1)

Anonymous Referee #1, 28 Spe 2024

The authors analyzed and presented the net ecosystem exchange (NEE) and the number concentration of negative intermediate ions ($N_{neg}$) measurements across sites in different ecosystems. They introduced the novel framework "CarbonSink + Potential" to highlight the importance of boreal ecosystems in the climate system. This framework offers an interesting and new perspective on how boreal ecosystems directly absorb $CO_2$ and indirectly influence the radiation balance, thereby mitigating global warming and climate change.

We thank the reviewer for the positive and constructive evaluation of our manuscript. All comments are addressed below and greatly helped to improve our manuscript.

We would like to make a note that we have corrected the measurement period at Haltiala croplands to '06/2021-10/2022' from '01/2022-12/2022'. All available data for both NAIS and $CO_2$ fluxes at Haltiala cropland are from 2021/06-2022/10 so far. We modified the text and the code so now it is the right date range. After revision, the midday $CO_2$ uptake rate and $N_{neg}$ in summer in Haltiala cropland decreased from 19.69 µmol m$^{-2}$ s$^{-1}$ and 3.08 cm$^{-3}$ to 10.42 µmol m$^{-2}$ s$^{-1}$ and 2.66 # cm$^{-3}$, respectively. The related figures and figures were all revised. However, the main conclusions remained without changes. Detailed responses to the comments can be found below.

This topic is both significant and relevant to the scope of Biogeoscience. While the paper is generally well-written, the analysis remains preliminary, and the main argument lacks clarity. I believe further analysis is needed to enhance the overall quality of manuscript. Here are some major comments I have:

1. I suggest that the authors change the way they present the data. Figures 2-4 and Figures 6-8 show two series of data for NEE and $N_{neg}$, respectively. I wonder why the authors presented the mean, 25th, and 50th percentiles for NEE (and the 50th and 75th percentiles for $N_{neg}$) separately in different panels. There are other effective options, such as box plots or violin plots. Additionally, could you combine the results for the different ecosystems? This could make it easier to see the differences between the ecosystems.

Reply: Thank you for the suggestion. In box plots (see examples below), the information is clustered together, making it too messy to include all ecosystems also in the same figure. As the main aim is to present the potential of $CO_2$ uptake (50th and 25th percentile) and aerosol production (50th and 75th percentile), we would prefer the original way of presenting the data. However, if the editor or reviewers insist, we can change the way of presenting the results.

[Figure]

Figure example1. The diurnal plot of the NEE in summer. The circles are mean values and the lines in the boxes are median values.

It is an interesting approach to consider the role of terrestrial ecosystems as direct carbon sinks and indirect sources of new particle precursors and aerosols. However, the comparison conducted in this study did not integrate these two concepts very well. For example, a recent study by Weber et al. (2024) illustrated this integration effectively. I wonder if the authors could provide an estimation of the relative importance of these two concepts.

Reply: The relative importance of $CO_2$ uptake and aerosol production can be compared by their contribution to radiative forcing, as done in the study by Weber et al., (2024). However, the process of aerosol growth, the radiation scattering effect, and aerosol-cloud interactions take place on a regional scale, whereas we aim to emphasize here the ecosystem-scale potentials of $CO_2$ uptake and aerosol production. We do this by utilizing the datasets of negative ions in specific size range, 2-2.3 nm. This is novel in our work and allows us to quantify ecosystems' climate cooling potential regarding aerosol production with a simplified method. How the ecosystem-scale $CO_2$ uptake and aerosol production impact the regional climate remains a topic to be addressed in a follow-up study.

2. Most of the analysis in this study is based on the diurnal cycle, but it lacks depth in data interpretation. I suggest conducting further analysis to provide more insights. For example, different ecosystems exhibit distinct terpenoid emission patterns. Many boreal needleleaf

forest ecosystems are dominated by monoterpenes (Boy et al., 2022), which are important precursors for particles. However, the fen at the Siikaneva site also has high isoprene emissions (Vettikkat et al., 2023), which could suppress the formation of new particles (Kiendler-Scharr et al., 2009). In addition, Vettikkat et al. (2023) reported high temperature sensitivity of terpenoids. I noticed that meteorological data was mentioned in Section 2.2 of the paper, but I did not see any related analysis. Could the authors incorporate additional analysis using meteorological data? For instance, how does $N_{neg}$ respond to temperature changes? How do different vegetation components or types affect the NEE and $N_{neg}$?

Reply: It is true that different ecosystems may exhibit distinct terpenoid emission patterns and show different responses to temperature and radiation, which can further influence aerosol production. However, the responses of $N_{neg}$ (aerosol production) and NEE to air temperature did not present a uniform trend and it is difficult to tell how different ecosystems respond differently to air temperature changes.

For example, when the PPFD is between 400 and 800 $\mu$mol m$^{-2}$ s$^{-1}$ in summer (figures below), there is a slight decreasing trend of $N_{neg}$ with increasing air temperature for sites other than Hyytiälä forest, which may be related to the cluster stability. In the case of NEE, all forest sites showed an increasing trend with air temperature, in contrast to the Siikaneva peatland. However, all the correlations were weak (R< 0.3). Air temperature can both increase the rates of respiration and photosynthesis, which makes it quite site-specific whether NEE (net $CO_2$ flux) will increase or decrease with air temperature. For $N_{neg}$, other factors, such as $H_2SO_4$ concentration, can distinctly impact the clustering formation. As the plots did not help to address our main research questions, we are not including them in the manuscript.

[Figure]

Figure example2: The responses of $N_{neg}$ and NEE to air temperature in summer when the photosynthetic photon flux density (PPFD) is between 400 and 800 µmol m$^{-2}$ s$^{-1}$. Half-hourly mean data are presented. The boxes are distributions of $N_{neg}$ and NEE in each air temperature bins of 5-10, 10-15, 15-20, 20-25°C.

We now have revised Figure 9 and added two paragraphs (lines 396-426), briefly reasoning why we see different NEE and $N_{neg}$ across the studied ecosystems. This included the analysis of air temperature and radiation:

"Multiple factors can cause the difference in NEE and $N_{neg}$ across the sites despite the similar seasonal and diurnal variation patterns. The $CO_2$ uptake rate at midday in summer increased with an increasing air temperature in both studied forests and agricultural fields (Figure 9). Moreover, the $CO_2$ uptake rate at midday in summer increased with LAI across the studied forest ecosystems (Table 1 and Figure S9). As F-RAN was selectively harvested (Section 2.3), the leaf area was decreased, which can result in a lower $CO_2$ uptake rate than other forests under similar air temperature and PPFD. Additionally, the peat soil at F-JAR and F-RAN can induce higher respiration (Figure 2). Hence, even though the LAI and air temperature at F-JAR were 23% and 10% higher than that in F-HYY, respectively, the NEE at F-JAR was only 4% lower than that at F-HYY. In the agricultural fields, the LAI

and air temperature were comparable or higher than that in the forests, which may explain the high momentary $CO_2$ uptake rate at summer midday in the agricultural fields.

In the case of $N_{neg}$, the precursor of aerosol production largely influences $N_{neg}$. The trends of $N_{neg}$ varying with air temperature and radiation were not evident (Figures 9 and S9). $H_2SO_4$ formation can drive the nucleation process and is influenced by the sulphur dioxide concentration and radiation. As the garden area and agricultural fields in this study are located in or nearby cities, the $SO_2$ concentration there may be enhanced due to the anthropogenic pollution and its long-range transport. Also, the terpene emissions can initiate NPF, which has been observed in Siikaneva peatland and led to stronger NPF there than that in F-HYY (Junninen et al., 2022; Huang et al., 2024). However, these events were reported to occur mostly in the late evening. Different plant species can emit different types of BVOCs (Guenther et al, 2012), e.g., monoterpenes are found dominant in coniferous forests and isoprene dominant in broadleaf forests. The oxidation products of monoterpenes can enhance aerosol formation and growth (Rose et al., 2018), while isoprene has been reported to inhibit new particle formation (Kiendler-Scharr et al., 2009). As birch species are mixed with coniferous species in F-JAR, the possibly higher isoprene emission than in the other three predominantly coniferous forests may partially explain the lower $N_{neg}$ in F-JAR. Moreover, the enhanced $NH_3$ in agricultural fields can play a synergistic role with both $H_2SO_4$ and low volatile organic compounds in clustering (Dada et al., 2023), which may explain the generally high $N_{neg}$ in the three studied agricultural fields."

[Figure]

Figure 9. Comparison between median NEE, median negative intermediate ions at 2.0-2.3 nm, and median air temperature at midday in summer between the sites. The error bars are 10[th] and 25[th] percentile for NEE, 75[th] and 90[th] percentile for the negative intermediate ions, and 75[th] and 90[th] percentile for the air temperature at each site.

[Figure]

Figure S9. Comparison between median NEE, median negative intermediate ions at 2.0-2.3 nm, leaf area index, and median photosynthetic photo flux density (PPFD) at midday in summer between the sites. The error bars are 10th and 25th percentile for NEE, 75th and 90th percentile for the negative intermediate ions, and 75th and 90th percentile for PPFD at each site.

**Minor comments:**

Line 180: The data periods differ among sites. Although the authors claimed that they would not discuss inter-annual variation, data from shorter periods, especially as short as one year, will still be affected by it, which may affect their diurnal cycles and comparisons with other sites. I think the authors should aware this and demonstrate the potential impact of inter-annual variation on their analysis.

Reply: We now have added the standard deviation of NEE and $N_{neg}$ at summer midday for all measurement years in Table 2 and discussed the potential impact of inter-annual variation on the result in lines 386-395 "It should be noted that only 1 year of data were applied in the stations with

newly established atmospheric measurement, i.e., C-TVA and A-VII, although measurements continue. The inter-annual variation of NEE has been widely observed in many ecosystems, e.g., F-HYY (Neefjes et al., 2022) and A-QVI (Heimsch et al., 2021), possibly due to inter-annual change in temperature and precipitation. In the reported year in C-TVA and A-VII, the air temperature was higher than average in years 2015-2020 (Finnish Meteorological Institute; Figure S8). Since a higher air temperature can simultaneously increase the rates of respiration and photosynthesis in an ecosystem, the influence of an increased air temperature on the net $CO_2$ flux, i.e., NEE, is quite site-specific. More observation years are needed to reduce the estimation errors of NEE. Compared with NEE, the $N_{neg}$ at summer midday was relatively stable across different years (Table 2). Hence the measured $N_{neg}$ in the reported year is likely representative of local aerosol production at the site."

Line 400: I don't understand the purpose of Figure 9. The error bars represent different percentiles on the x and y axes, and the meaning of the dots is not explained (are they means? medians?). In addition, I expected the authors to discuss the relationship between NEE and $N_{neg}$, but this scatter plot does not seem to address that. It is more like putting data together.

Reply: Line 413 for the figure caption is revised "Figure 9. Comparison between the median NEE, median negative intermediate ions at 2.0-2.3 nm, and median air temperature at midday in summer between the sites. The error bars are the 10th and 25th percentiles for NEE, the 75th and 90th percentiles for the negative intermediate ions, and the 75th and 90th percentiles for the air temperature at each site". The Figure 9 has been revised and we now briefly discuss the factors that can cause the observed differences in NEE and $N_{neg}$ across the ecosystems (see reply above). The main scope is to present the potential of different ecosystems influencing the $CO_2$ uptake and local aerosol production. By clustering similar ecosystems, Figure 9 clearly demonstrates that the differences in NEE and $N_{neg}$ are associated quite strongly with the type of an ecosystem, and not only with meteorological drivers.

The direct connection between NEE and $N_{neg}$ was weak within sites (see example picture below for Hyytiälä forest in summer, when the photosynthetic photon flux density is higher than 600 µmol m$^{-2}$ s$^{-1}$. The 10th, 25th, 50th, 75th, and 90th percentile of $N_{neg}$ in each NEE bin were plotted). Explaining the correlation between NEE and $N_{neg}$ is out of scope of the present manuscript and will be investigated in follow-up studies.

[Figure]

Figure example3. The correlation between NEE and $N_{neg}$ at Hyytiälä forest in summer when PPFD is above 600 µmol m⁻² s⁻¹. Half-hour mean data are used.

**Reference**

Boy, M., Zhou, P., Kurtén, T., Chen, D., Xavier, C., Clusius, P., Roldin, P., Baykara, M., Pichelstorfer, L., Foreback, B., Bäck, J., Petäjä, T., Makkonen, R., Kerminen, V.-M., Pihlatie, M., Aalto, J., and Kulmala, M.: Positive feedback mechanism between biogenic volatile organic compounds and the methane lifetime in future climates, npj Climate and Atmospheric Science, 5, 72, 10.1038/s41612-022-00292-0, 2022.

Kiendler-Scharr, A., Wildt, J., Maso, M. D., Hohaus, T., Kleist, E., Mentel, T. F., Tillmann, R., Uerlings, R., Schurr, U., and Wahner, A.: New particle formation in forests inhibited by isoprene emissions, Nature, 461, 381-384, 10.1038/nature08292, 2009.

Vettikkat, L., Miettinen, P., Buchholz, A., Rantala, P., Yu, H., Schallhart, S., Petäjä, T., Seco, R., Männistö, E., Kulmala, M., Tuittila, E. S., Guenther, A. B., and Schobesberger, S.: High emission rates and strong temperature response make boreal wetlands a large source of isoprene and terpenes, Atmos. Chem. Phys., 23, 2683-2698, 10.5194/acp-23-2683-2023, 2023.

Weber, J., King, J. A., Abraham, N. L., Grosvenor, D. P., Smith, C. J., Shin, Y. M., Lawrence, P., Roe, S., Beerling, D. J., and Martin, M. V.: Chemistry-albedo feedbacks offset up to a third of forestation's CO2 removal benefits, Science, 383, 860-864, 10.1126/science.adg6196, 2024.

---

## Author Comment (AC2)

Potential of carbon uptake and local aerosol production in boreal and hemi-boreal ecosystems across Finland and in Estonia

Piaopiao Ke et al.,

This manuscript explained and researched the $CO_2$ uptake and local aerosol production at different types of stations in Finland. Authors analysed each NEE and $N_{neg}$ characteristics at each station and compare between them. Introduction is very persuasive and interesting why this study is necessary. However, the way to describe the result of experiments is not kind to the readers. Result was simple and it seems like lack of discussion. I also wonder whether the conclusion gives the answer to the question in the Introduction section. Hopefully authors read the manuscript carefully with reader's view and revise/describe them explicitly.

We are grateful for insightful comments and suggestions provided by the reviewer. All comments are addressed below and have significantly improved our manuscript.

We would like to make a note that we have corrected the measurement period at Haltiala croplands to '06/2021-10/2022' from '01/2022-12/2022'. All available data for both NAIS and $CO_2$ fluxes at Haltiala cropland are from 2021/06-2022/10 so far. We modified the text and the code following the corrected date range. After revision, the midday $CO_2$ uptake rate and $N_{neg}$ in summer in Haltiala cropland decreased from 19.69 µmol m$^{-2}$ s$^{-1}$ and 3.08 # cm$^{-3}$ to 10.42 µmol m$^{-2}$ s$^{-1}$ and 2.66 # cm$^{-3}$, respectively. The related figures and tables were all revised. However, the main conclusions remained without changes. Detailed responses to the comments can be found below.

General comments

1. Title mentioned 'boreal and hemi-boreal ecosystem' but all manuscripts are not linked to boreal and hemi-boreal region. Authors did not explain which site is belonging to boreal or hemi-boreal. According to the category in Table 1, there are forest, agricultural, peatland, urban garden and coastal area. It was hard to match to the title. Authors should consider how to make this manuscript well match to the title and more interesting.

   Reply: We have now added references assigning climate zones to these sites, in line 117 "F-JAR, C-TVA, and A-QVI belong to hemi-boreal ecosystems, while the other ecosystems are boreal (Mäki et al., 2022)". In Table 1, the climate zone of each station is added.

2. Authors did not make station ID and mentioned the station name directly (Table 1). But readers are not familiar with Finish and hard to follow up what type of station with only station's names. Hope authors reconsider the way to explain station' name.

   Reply: We now have added the station ID for each site and revised the name throughout the manuscript. We hope this is clear now.

| Sites (Site ID) | | Location | Selected period | Mean air temperature (°C) | Rainfall (mm/yr) | Dominant plant species | Peak LAI | Climate Zone |
|---|---|---|---|---|---|---|---|---|
| Forest | Hyytiälä, SMEAR II (F-HYY) | 61°51' N, 24°17' E | 11/2009-12/2022 | 4.8 | 709[1] | Scots pine and Norway spruce | 4.6 | Boreal |
| | Värriö, SMEAR I (F-VAR) | 67°46' N, 29°35' E | 3/2019-12/2022 | 0.4 | 601[2] | Scots pine | 3.2 | Boreal |
| | Ränskälänkorpi (F-RAN) | 61°10' N, 25°16' E | 4/2021-12/2022 | 5.4 | 600[3] | Norway spruce, Scots pine, downy birch | ---- | Boreal |
| | Järvselja, SMEAR Estonia (F-JAR) | 58°16' N, 27°16' E | 10/2016-12/2020 | 6.8 | 500-750[4] | Birch species, Scots pine, Norway spruce | 6 | Hemi-boreal |
| Agricultural fields | Haltiala, SMEAR Agri (A-HAL) | 60°16' N, 24°57' E | 6/2021-10/2022 | 6.5 | 700[5] | Oat | 5.5 | Boreal |
| | Qvidja (A-QVI) | 60°18' N, 22°24' E | 12/2018-8/2022 | 7.0 | 679[6] | Timothy, meadow fescue | 6.2 | Hemi-boreal |
| | Viikki, SMEAR Agri (A-VII) | 60°13' N, 25°01' E | 7/2022-6/2023 | 6.5 | 792[5] | Timothy (2022), Barley (2023) | 5.2 | Boreal |
| Peatland | Siikaneva, SMEAR II (P-SII) | 61°50' N, 24°12' E | 11/2019-12/2022 | 5.0 | 710[7] | Moss and sedges | 0.6 | Boreal |
| Urban garden | Kumpula, SMEAR III (G-KUM) | 60°12' N, 24°58' E | 5/2016-12/2022 | 6.3[5] | 731[5] | Mixed | ------ | Boreal |

| Coastal area | Tvärmin ne (C-TVA) | 59°51' N, 23°15' E | 6/2022-8/2023 | 7.2[5] | 639[5] | Seagrass and seaweed | ----- | Hemi-boreal |
|---|---|---|---|---|---|---|---|---|

3. In many paragraphs, authors did not suggest table/figure # but also values. For example, *L367-369: The momentary net $CO_2$ uptake rate at midday in summer was highest in agricultural fields, followed by the forests. The urban garden in this study displayed distinct net $CO_2$ uptake, lower than the forests and higher than the open peatland*. Next to the highest, it needs value. Lower than or higher than, authors can suggest certain values. Without certain values, all explanations such as "lower than, higher than, highest, similar" are very boring and insufficient to deliver what authors would like to say.

Reply: We have now revised the descriptions throughout the manuscript and added exact values for the differences, such as in lines 371-378: "The agricultural fields exhibited a 47% higher $N_{neg}$ compared to the studied forests. In contrast, the open peatland (P-SII) had a 23% lower $N_{neg}$ than F-HYY but 15-46% higher $N_{neg}$ than the other forests. The $N_{neg}$ at the coastal area was the lowest. The momentary net $CO_2$ uptake rate at midday in summer was highest in agricultural fields, followed by the forests. The urban garden in this study displayed distinct net $CO_2$ uptake, 37% lower than the forests and ~2 times that in the open peatland. The coastal area at midday in summer was a very weak $CO_2$ sink. In the urban garden area in Kumpula, median $N_{neg}$ was double of that in Hyytiälä forest, while the median NEE only reached 63% of that in Hyytiälä forest". For general descriptions, such as in Line 271, "For median values in summer, $N_{neg}$ was found to be the highest in the urban garden, followed by the agricultural fields (Figure 9)", we would like to follow the present style to keep the manuscript concise.

4. Hope authors can describe the sufficient reason after the result, for example, in section 3.3. This is scientific paper not a report.

Reply: In this study, we aimed to present the different potential of $CO_2$ uptake and aerosol production across different ecosystems; hence we reported directly measured $CO_2$ fluxes and local aerosol production, indicated by the negative ions at 2-2.3 nm. We focus on the comparison of potentials and have now briefly investigated the reasons that could cause the difference in NEE and $N_{neg}$ between the ecosystems. We have revised Figure 9 and added a corresponding discussion in two paragraphs in Section 3.3 (Line 396-426):

"Multiple factors can cause the difference in NEE and $N_{neg}$ across the sites despite the similar seasonal and diurnal variation patterns. The $CO_2$ uptake rate at midday in summer increased with an increasing air temperature in both studied forests and agricultural fields (Figure 9). Moreover, the $CO_2$ uptake rate at midday in summer increased with LAI across the studied forest ecosystems (Table 1 and Figure S9). As F-RAN was selectively harvested (Section 2.3), the leaf area was decreased, which can result in a lower $CO_2$ uptake rate than other forests under similar air temperature and PPFD. Additionally, the peat soil at F-JAR and F-RAN can induce higher respiration (Figure 2). Hence, even though the LAI and air temperature at F-JAR were 23% and

10% higher than that in F-HYY, respectively, the NEE at F-JAR was only 4% lower than that at F-HYY. In the agricultural fields, the LAI and air temperature were comparable or higher than that in the forests, which may explain the high momentary $CO_2$ uptake rate at summer midday in the agricultural fields.

In the case of $N_{neg}$, the precursor of aerosol production largely influences $N_{neg}$. The trends of $N_{neg}$ varying with air temperature and radiation were not evident (Figures 9 and S9). $H_2SO_4$ formation can drive the nucleation process and is influenced by the sulphur dioxide concentration and radiation. As the garden area and agricultural fields in this study are located in or nearby cities, the $SO_2$ concentration there may be enhanced due to the anthropogenic pollution and its long-range transport. Also, the terpene emissions can initiate NPF, which has been observed in P-SII and led to stronger NPF there than that in F-HYY (Junninen et al., 2022; Huang et al., 2024). However, these events were reported to occur mostly in the late evening. Different plant species can emit different types of BVOCs (Guenther et al, 2012), e.g., monoterpenes are found dominant in coniferous forests and isoprene dominant in broadleaf forests. The oxidation products of monoterpenes can enhance aerosol formation and growth (Rose et al., 2018), while isoprene has been reported to inhibit new particle formation (Kiendler-Scharr et al., 2009). As birch species are mixed with coniferous species in F-JAR, the possibly higher isoprene emission than in the other three predominantly coniferous forests may partially explain the lower $N_{neg}$ in F-JAR. Moreover, the enhanced $NH_3$ in agricultural fields can play a synergistic role with both $H_2SO_4$ and low volatile organic compounds in clustering (Dada et al., 2023), which may explain the generally high $N_{neg}$ in the three studied agricultural fields."

[Figure]

Figure 9. Comparison between median NEE, median negative intermediate ions at 2.0-2.3 nm, and median air temperature at midday in summer between the sites. The error bars are 10th and 25th percentile for NEE, 75th and 90th percentile for the negative intermediate ions, and 75th and 90th percentile for the air temperature at each site.

[Figure]

Figure S9. Comparison between median NEE, median negative intermediate ions at 2.0-2.3 nm, leaf area index, and median photosynthetic photo flux density (PPFD) at midday in summer between the sites. The error bars are $10^{th}$ and $25^{th}$ percentile for NEE, $75^{th}$ and $90^{th}$ percentile for the negative intermediate ions, and $75^{th}$ and $90^{th}$ percentile for PPFD at each site.

Minor comments

1. L58: Räty et al., 2023 was not on the reference list.

   Reply: It is now added to the list.

2. L86: long-term datasets. What's the definition of 'long-term' here? Only long-term datasets are from Hyytiälä station and others are less than a decade.

   Reply: It is revised to "This study utilized 1 to 10 year-long datasets of intermediate ion concentrations."

3. Table 1: Between the category (such as Forest and Agricultural) it would be good to have a line.

   Reply: The lines are added now.

4. Table 1: Hope station name can be reconsider. Especially Hyytiälä station is 'reference station', used everywhere but in the forest category. Author can name stations with their characteristic.

   Reply: The site ID is now added (see response to the major comment 2) and used throughout the manuscript. We took Hyytiälä forest as a reference station as it is one typical ecosystem type in Finland and has the longest record of data and endures relatively little anthropogenic interference. Hence, we still kept the Hyytiälä site in forest category.

5. Table 1: Location. Hope author can add height of each station.

   Reply: We added this information in Table S1. We have added in line 145 "The inlets for all the NAIS in the studied sites are 1-2 m high above the ground." and in line 170 "The fluxes were measured above the ecosystem canopies and below 30 m. The detailed measurement height for each site is listed in Table S1.

   L133: *at same heights*. Please add values.

   Reply: The height information is included in Table S1. As the measurement heights (1-70 m) vary between the stations, dependent on the ecosystem canopy height, we referred to Table S1 in the text.

6. L137: *Diameter range*. Does it mean particle size?

   Reply: Yes, it describes the range of particle size in diameter.

7. L140: *The data were cleaned*. What does it mean? Does it mean 'filtered' or 'selected'?

   Reply: This is now deleted as it is the same meaning of 'quality-checked'. "The data were quality-checked, considering e.g. the potential interference of rainfall and snow events on the measurements (Manninen et al., 2016)."

8. L142: *total particle concentration*. Is it mass concentrations or number concentrations or others?

   Reply: It is a number concentration. It is revised as "The ion and total particle number concentration" in the text.

9. L151: *their footprints are constrained within the ecosystem scale*. Don't authors think that it can be differed according to the height of station?

Reply: Indeed, the footprints of both eddy covariance and $N_{neg}$ are impacted by the height of the station. This sentence is revised as "their footprints are constrained within the ecosystem scale when measured at a height between 1 and 70 m (Section 2.2 and Table S1)." The forest canopy is below 30 m and even shorter for agricultural fields and peatlands. The footprint for eddy covariance (Kljun et al., 2015) and $N_{neg}$ (Tuovinen et al., 2024) are within the studied ecosystems.

10. L153: *outside the active hours of the ecosystem, were taken as the background concentration at each site*. This sentence is conflict against L352-357.

Reply: The ecosystem at nighttime is assumed to be relatively inactive with no photosynthesis and low BVOC emission. The nighttime clustering explained in L352-357 is more likely a consequence of meteorological conditions associated with atmospheric chemical reactions. To clarify the difference between drivers for daytime and nighttime clustering, the sentence is revised as "However, these negative ions clustered at nighttime are likely unable to grow >3 nm in diameter (Mazon et al., 2016)."

11. L194: *tower*. Hope to add the tower height.

Reply: The tower height can differ largely with the measurement station. Here we added the measurement height in case of confusion. We revised the text as "The NAIS equipment was positioned in the border between the control and clear-out, ~230 m east from the eddy covariance tower (measurement height of 29 m)".

12. L203: *interference*. What kind of interference?

Reply: There is an experimental area (for different cutting heights) between 30° and 140° of the eddy covariance mast in the agricultural grassland. The different management in the experimental area may influence the measured $CO_2$ uptake and $N_{neg}$. This sentence was revised to avoid confusion "For the Qvidja site, NAIS and eddy covariance data from wind directions between 30° and 140° were discarded due to another experimental plot located in that part of the field (Heimsch et al., 2021)".

13. L205: *fields*. What is the possible source to affect on data?

Reply: Revised as "Similarly, at the Viikki site, only measurements from wind direction between 145° and 245° were included in the analysis to avoid data from other nearby fields with different vegetation and management activities." Vegetation and management activities, such as fertilization, may impact the $CO_2$ uptake, emission of BVOCs and $NH_3$ concentration.

14. L226: *In the other seasons, the urban garden area was a net source of $CO_2$ most of the time, similar to the results previously reported for the years 2006-2010 from the same site (Järvi et al., 2012)*. Most of section did not explain the reason. Why is that?

Reply: This part is now revised (L230): "There are residential buildings and traffic within the eddy covariance measurement footprint in G-KUM. The $CO_2$ emissions from the residential buildings, traffic and bare soil outweighed photosynthetic uptake of $CO_2$ except during summer daytime".

15. Figure 2-4: Why did authors add 50[th] and 25[th] graph? It would better to add median with standard deviation to see the variation of the values since experimental periods are different at each station.

Reply: The 50th percentile (median) of the NEE described the average status of $CO_2$ fluxes. For the 25th percentile of $CO_2$ fluxes, they corresponded to the conditions when the ecosystem is very active. In the case of $N_{neg}$, 75[th] percentile can be a sign of new particle formation event (Aliaga et al., 2023). We want a uniform presentation of NEE and $N_{neg}$ in the manuscript. Also, as we aimed to emphasize the $CO_2$ uptake and local aerosol production potentials, we prefer keeping the original way of presenting the data.

16. L237: *In the case of agricultural fields in summer (Figure 3), the Haltiala site had higher momentary net $CO_2$ uptake than the other two agricultural sites*. Why don't authors add specific value for 'higher than' precisely?

Reply: It is revised to "In the case of agricultural fields in summer (Figure 3), the A-HAL and A-VII croplands had 2-5 μmol m$^{-2}$ s$^{-1}$ (for midday median values) higher momentary net $CO_2$ uptake rate than the other agricultural grassland." The similar issue, i.e., the specific values for "higher than" or "lower than", in the whole manuscript have been revised.

17. L239: *Qvidja was a $CO_2$ sink during daytime with a similar uptake rate to the Hyytiälä forest.* Add specific value and figure #.

Reply: This line is revised as "A-QVI was a $CO_2$ sink during daytime with a comparable uptake rate to the F-HYY (ranging between 0 and 4 μmol m$^{-2}$ s$^{-1}$)."

18. L240: *The different plant species (Table 1) and management activities between the agricultural fields likely caused the differences in their seasonal $CO_2$ fluxes*. This is very confusing to readers. In table 1, authors mentioned of plants type but not management activities. I confirmed that it was described in the section 2.1, it would be good to add table 1.

Reply: We now added the description of management activities in the three agricultural fields in Section 2.3 (line 143) "A-QVI was harvested in June and August, A-VII was harvested twice in August during the reported period, and A-HAL was harvested once around the end of August during the measurement periods. The sowing (over-seeding for A-QVI and only in 2022) and first fertilization in the year usually takes place at the end of spring." And following line 240 "The different plant species (Table 1) and management activities between the agricultural fields likely caused the differences in

their seasonal $CO_2$ fluxes. During the measurement period, perennial plants were grown in A-QVI, while the growth of the annual crops in A-HAL and A-VII relied on the sowing and fertilization date, normally at the end of spring. This may explain the springtime $CO_2$ emission in A-HAL and A-VII. In the summer, the A-HAL and A-VII was harvested only in August, while A-QVI was harvested in June and August separately, which may explain the higher CO2 uptake rate in A-HAL and A-VII."

19. L294: *The median values of $N_{neg}$ in the daytime in spring were higher than those in the Haltiala and Viikki croplands, Siikaneva peatland, and Kumpula urban garden area*. I cannot understand what this sentence means.

    Reply: Revised as follows "The daytime median values of $N_{neg}$ were higher in spring than that in summer at A-HAL and A-VII, P-SII, and G-KUM. At the other sites, the median values in summer were higher than those in spring".

20. L296: *summer median values were higher*. Please finalize the sentence. Higher than what?

    Reply: Revised "At the other sites, the median values in summer were higher than those in spring".

21. L338: *The application of fertilizers in agricultural fields is known to remarkably increase the atmospheric concentration of ammonia ($NH_3$)*. This sentence seems to be generalized to all stations. Please add specific type of regions.

    Reply: It is revised to "The application of fertilizers is known to remarkably increase the atmospheric concentration of ammonia ($NH_3$) in agricultural fields, e.g., observed in A-QVI (Olin et al., 2022)".

22. Figure 9: I cannot understand the meaning of bar in Figure 9.

    Reply: Line 413 for the figure caption is revised "Figure 9. Comparison between the median NEE, median negative intermediate ions at 2.0-2.3 nm, and median air temperature at midday in summer between the sites. The error bars are the 10th and 25th percentiles for NEE, the 75th and 90th percentiles for the negative intermediate ions, and the 75th and 90th percentiles for the air temperature at each site". The main scope is to present the potential of different ecosystems influencing the $CO_2$ uptake and local aerosol production. The $25^{th}$ and $75^{th}$ percentile for NEE and $N_{neg}$ were presented, respectively, to show the higher range of ecosystem $CO_2$ uptake rate and local aerosol production.

23. Table 2: There is no explanation of 'c' (superscript) and there is no 'b' in the table.

    Reply: The 'c' is revised to be 'b' in Table 2.

24. L409: Summary and conclusions or only summary?

25. L416: What are 10 stations belonging to hemi-boreal station? Why did not author define 'hemi-boreal' and 'boreal' in the manuscript?

    Reply: This has been addressed above.

26. L421-423: Why didn't authors add the reasons?

    Reply: This line is revised as "A distinct $CO_2$ uptake in the urban garden at midday in summer was observed, due to the strong photosynthesis of vegetation inside. The uptake rate was 37% lower than that in F-HYY but ~2 times of that observed in the open peatland". We also simply explained why agricultural fields presented generally high $CO_2$ uptake and aerosol production in Line 490 "The results showed that the agricultural fields had similar or even 15% higher $CO_2$ uptake potentials compared to the boreal Hyytiälä forest (F-HYY) during the summer, possibly due to the high leaf area index and air temperature in the agricultural fields." and line 500 "In agricultural fields, the synergetic role of $NH_3$, $H_2SO_4$, and low volatile organic compounds originating from BVOC oxidation may play a synergistic role in clustering and induce a high $N_{neg}$ comparing with other ecosystem types.".

27. L431: What's the difference between reference station and background station? Please add the definition.

    Reply: The background site in this context means the sites that receive little human interference. Here we actually did not need the concept of background sites, and the text is revised as "Note that the urban garden and agricultural sites in Helsinki might be more influenced by air pollution compared to the forests and open peatland that received little anthropogenic interference and pollution".

Reference

Kljun, N., Calanca, P., Rotach, M. W., and Schmid, H. P.: A simple two-dimensional parameterisation for Flux Footprint Prediction (FFP), Geoscientific Model Development, 8, 3695-3713, https://doi.org/https://doi.org/10.5194/gmd-8-3695-2015, 2015.
Tuovinen, S., Lampilahti, J., Kerminen, V.-M., and Kulmala, M.: Intermediate ions as indicator for local new particle formation, Aerosol Research, 2, 93-105, https://doi.org/10.5194/ar-2-93-2024, 2024.

---

## Referee Report (RR1)

Potential of carbon uptake and local aerosol production in boreal and hemi-boreal ecosystems across Finland and in Estonia

Piaopiao Ke et al., 2025

General comments

This manuscript was well revised according to reviewer's comment. Authors tried to answer to the question reviewers raised and refine the manuscript. It becomes readable easily compared to the previous version. Very appreciated.

Despite author's great efforts, the conclusion, readers can get through this manuscript, still seems to be very simple. Unique part in this manuscript is authors used two different data set with $CO_2$ flux and aerosols. However, the conclusion might not include those data enough. It doesn't mean that experiment results are not enough. I believe all data and experimental setting are quite good enough.

Authors' conclusion is *"Overall, considering the large area of forests in Finland and Estonia, the forests in total have the largest potential of climate cooling when considering the $CO_2$ uptake and local new particle formation."*. There are three points. 1) large area 2) $CO_2$ uptake 3) local new particle formation. First, authors never discussed that land size is one of factors in the manuscript. It should be re-considered whether it is valuable to make a point with the land size here. Authors explained the agriculture field is comparable to forest for $CO_2$ uptake during summer and high $N_{neg}$ was observed. Based on this result, readers can assume that agricultural field can be an option for climate change policy. If not, authors should answer to the question, why forest is more important than agriculture fields with a new finding based on two different data. This is well known fact that forest $CO_2$ sink/summertime $CO_2$ sink is stronger than other areas (such as urban garden, agriculture and coastal site) and other seasons based on many of previous papers.

If the conclusion ends up with that forest is the best place for climate change policy with common knowledges, this manuscript cannot be valuable to be published.

Authors should re-consider and explain major points such as how all data sets are used to make conclusion (all data can be linked together) and what the new findings are here when two data sets are used. And then it can be considered to be published.

Minor comments

1. Reference station/reference data: It is still unclear to define Hyytiälä forest (F-HYY) as a reference site. When we think of a reference site (background site) for CO2 flux, it might be chosen by environments without any variation/fluctuations

like costal site (see Figure 4). It is hard to understand the reason to choose F-HYY as a reference site.

2. Height of each station: When seeing table S1, the instrument heights were quite different from each other. This can make a bias to analyse NEE when their values were compared to each other. Is it enough to explain that the height can be represented each site characteristic?

3. Figure 7: no explanation of a) in the caption.

4. Figure 9: If the error bars mean $10^{th}$ and $25^{th}$ percentile for NEE, is it necessary to + and – value?  It is hard to understand of the graph. Also, there is no explanation of a) to c) in the caption and even in the manuscript.

---

## Author Response (AR2)

Potential of carbon uptake and local aerosol production in boreal and hemi-boreal ecosystems across Finland and in Estonia

Piaopiao Ke et al., 2025

General comments

This manuscript was well revised according to reviewer's comment. Authors tried to answer to the question reviewers raised and refine the manuscript. It becomes readable easily compared to the previous version. Very appreciated.

Despite author's great efforts, the conclusion, readers can get through this manuscript, still seems to be very simple. Unique part in this manuscript is authors used two different data set with $CO_2$ flux and aerosols. However, the conclusion might not include those data enough. It doesn't mean that experiment results are not enough. I believe all data and experimental setting are quite good enough.

Authors' conclusion is *"Overall, considering the large area of forests in Finland and Estonia, the forests in total have the largest potential of climate cooling when considering the $CO_2$ uptake and local new particle formation.".* There are three points. 1) large area 2) $CO_2$ uptake 3) local new particle formation. First, authors never discussed that land size is one of factors in the manuscript. It should be re-considered whether it is valuable to make a point with the land size here. Authors explained the agriculture field is comparable to forest for $CO_2$ uptake during summer and high $N_{neg}$ was observed. Based on this result, readers can assume that agricultural field can be an option for climate change policy. If not, authors should answer to the question, why forest is more important than agriculture fields with a new finding based on two different data. This is well known fact that forest $CO_2$ sink/summertime $CO_2$ sink is stronger than other areas (such as urban garden, agriculture and coastal site) and other seasons based on many of previous papers.

If the conclusion ends up with that forest is the best place for climate change policy with common knowledges, this manuscript cannot be valuable to be published.

Authors should re-consider and explain major points such as how all data sets are used to make conclusion (all data can be linked together) and what the new findings are here when two data sets are used. And then it can be considered to be published.

We appreciate your careful and thorough reviewing of the manuscript very much. We agree that the conclusion should be revised to be more explicit. Both the land size and single site' potential are important in our study. We now have revised the Section 3.3 and Conclusion to address the following points:

- Momentary CO2 uptake rate and local aerosol production: We have clarified that "our results showed that agricultural fields have highest potential to contribute to momentary $CO_2$ uptake and aerosol formation, affected by their vegetation and management practices. However, carbon inputs from fertilization and removal through

harvested biomass in agricultural fields, which were not considered in our study, can lead to net carbon emissions in the annual carbon budgets (Heimsch et al., 2021; and references therein)" (Lines 442-446). This is the most unexpected results from our study, indicating that agricultural sites have both high momentary $CO_2$ uptake and local aerosol production. However, it should be noted that the momentary $CO_2$ uptake is only one part of the whole carbon budget of an agricultural field.

- Large area: We have added a detailed discussion on the significance of land size in the context of discussing different ecosystems' cooling potential via $CO_2$ uptake and aerosol production, "Moreover, forests are the dominant landscape in Finland, covering ~9 times the area of agricultural fields (Table 2). Considering their large area, boreal forests in Finland are likely the largest contributor of climate cooling when considering the $CO_2$ uptake and local new particle formation" (Lines 446-469). In the Conclusion, it was revised to "Overall, considering the large area of forests in Finland and Estonia, the forests in total are the largest contributors to climate cooling in terms of their CO2 uptake and local new particle formation" (Line 526). It is important to point out that in present days, forests are still the largest contributor to climate cooling in Finland, providing a comprehensive conclusion of the comparison between different ecosystems.

The direct connection between NEE and $N_{neg}$ was weak at each site (see example picture below for Hyytiälä forest in summer, when the photosynthetic photon flux density is higher than 600 $\mu$mol m$^{-2}$ s$^{-1}$. The 10th, 25th, 50th, 75th, and 90th percentile of $N_{neg}$ in each NEE bin were plotted). This suggests that the momentary carbon uptake and aerosol forming potential of the ecosystems investigated here are weakly coupled to each other, despite the theoretical connection. However, it is important to recognize that both carbon uptake and aerosol formation are essential processes for ecosystems to contribute to climate cooling. Our results compared the carbon uptake and aerosol formation potential across different ecosystems. Quantifying the potential connection between the carbon uptake and aerosol forming potential of ecosystems requires a deeper analysis with careful consideration of the meteorological and other factors influencing these two processes, which is beyond the scope of the present manuscript and will be investigated in follow-up studies.

[Figure]

Figure example1. The correlation between NEE and $N_{neg}$ at Hyytiälä forest in summer. Half-hour mean data are used.

Another pathway to link the two datasets in this study can be comparing their relative contributions to radiative forcing, as done in the study by Weber et al. (2024). However, the process of aerosol growth, the radiation scattering effect, and aerosol-cloud interactions take place on a regional scale, whereas we aim to emphasize here the ecosystem-scale potentials of $CO_2$ uptake and aerosol production. We do this by utilizing the datasets of negative ions in specific size range, 2-2.3 nm. This is novel in our work and allows us to quantify ecosystems' climate cooling potential regarding aerosol production with a simplified method. How the ecosystem-scale $CO_2$ uptake and aerosol production impact the regional climate remains a topic to be addressed in the future studies.

We have now provided a more comprehensive and nuanced conclusion. We hope these revisions address the reviewer's concerns well and the present manuscript meets the standard of publication.

Minor comments

1. Reference station/reference data: It is still unclear to define Hyytiälä forest (FHYY) as a reference site. When we think of a reference site (background site) for CO2 flux, it might be chosen by environments without any variation/fluctuations like costal site (see Figure 4). It is hard to understand the reason to choose F-HYY as a reference site. It should also be noted that here the "reference site" is not taken as a background information site to investigate the effects of some disturbances/management practices/pollutions. It is simply applied for easier comparison between each ecosystem. To make it clearer, we revised the text to "The value of NEE and $N_{neg}$ at the boreal forest in Hyytiälä (F-HYY) were used as references, to which NEE and $N_{neg}$ at all other sites were compared" (Line 393). Forests are the largest types of land cover in Finland (Table 2). Also, F-HYY has the longest record of data and endures relatively little anthropogenic interference (Line 194). The costal sites still presented seasonal changes in NEE and $N_{neg}$, although the seasonal changes were in a much smaller

magnitude than that in other ecosystems. And also, the dataset at that site was one-year long, and the inter-year change can drive evident variation in its NEE and local aerosol production values.

2. Height of each station: When seeing table S1, the instrument heights were quite different from each other. This can make a bias to analyse NEE when their values were compared to each other. Is it enough to explain that the height can be represented each site characteristic?

The measurement height for NEE is all above the canopy height of each terrestrial ecosystem, while the NAIS is all 1-2 m above ground at all studied sites (Line 147). To avoid confusion, we added canopy height to Table S1 now. The measurement height for eddy covariance at each station was carefully selected to represent the whole ecosystem (Järvi et al., 2012; Alekseychik et al., 2021; Heimsch et al., 2021; Laurila et al., 2021). At the height of 3 m, 80% of the footprint of the eddy covariance measurements covered an area with a radius 100 m (Alekseychik et al., 2021). The NEE at Qvidja, which has the lowest measurement height of NEE, 2.3 m, the data were discarded when the flux footprint was not sufficiently representative of the target grassland (Heimsch et al., 2021; Line 212). Hence the measured NEE at each station can sufficiently represent the site characteristics. Although the footprint at taller measurement heights covered a larger area, as long as the main area constrained for the ecosystem, the measured NEE can still represent the whole ecosystem.

3. Figure 7: no explanation of a) in the caption.

Revised to "Figure 7. The 50th (a) and 75th percentile (b) of negative intermediate ions ($N_{neg}$) at 2.0-2.3 nm at each hour and the daily fluctuations of $N_{neg}$ (c) for the agricultural fields in spring (MAM) and the corresponding 50th percentile, 75th percentile and normalized concentration for median values, (d), (e), (f), in summer (JJA), respectively".

4. Figure 9: If the error bars mean 10$^{th}$ and 25$^{th}$ percentile for NEE, is it necessary to + and – value? It is hard to understand of the graph. Also, there is no explanation of a) to c) in the caption and even in the manuscript.

The caption was revised now as "Figure 9. Comparison of NEE and negative intermediate ions at 2.0-2.3 nm (a), NEE and air temperature (b), and negative intermediate ions at 2.0-2.3 nm and air temperature (c) across different sites. The dots represent median values during summer midday (10:00-14:00). Error bars indicate the 10th and 25th percentiles for NEE, and the 75th and 90th percentiles for negative intermediate ions and air temperature, reflecting the $CO_2$ uptake rate and aerosol formation under optimal conditions". We focused on the potential of ecosystem $CO_2$ uptake and aerosol production, i.e., how these will be under optimal conditions. Hence, the 90% and 75% of $N_{neg}$ and 10% and 25% for NEE were kept. It is not a graph that is very direct, and we revised the caption to make it easier to understand.

Reference:

Alekseychik, P., Korrensalo, A., Mammarella, I., Launiainen, S., Tuittila, E.-S., Korpela, I., and Vesala, T.: Carbon balance of a Finnish bog: temporal variability and limiting factors based on 6 years of eddy-covariance data, Biogeosciences, 18, 4681-4704, https://doi.org/https://doi.org/10.5194/bg-18-4681-2021, 2021.

Heimsch, L., Lohila, A., Tuovinen, J.-P., Vekuri, H., Heinonsalo, J., Nevalainen, O., Korkiakoski, M., Liski, J., Laurila, T., and Kulmala, L.: Carbon dioxide fluxes and carbon balance of an agricultural grassland in southern Finland, Biogeosciences, 18, 3467-3483, https://doi.org/https://doi.org/10.5194/bg-18-3467-2021, 2021.

Järvi, L., Nordbo, A., Junninen, H., Riikonen, A., Moilanen, J., Nikinmaa, E., and Vesala, T.: Seasonal and annual variation of carbon dioxide surface fluxes in Helsinki, Finland, in 2006–2010, Atmos. Chem. Phys., 12, 8475-8489, https://doi.org/https://doi.org/10.5194/acp-12-8475-2012, 2012.

Laurila, T., Aurela, M., Hatakka, J., Hotanen, J.-P., Jauhiainen, J., Korkiakoski, M., Korpela, L., Koskinen, M., Laiho, R., Lehtonen, A., Alder, K., Linkosalmi, M., Salmon, A., Minkkinen, K., Mäkelä, T., Mäkiranta, P., Nieminen, M., Ojanen, P., Peltoniemi, M., Penttilä, T., Rainne, J., Rautakoski, H., Saarinen, M., Salovaara, P., Sarkkola, S., and Mäkipää, R.: Set-up and instrumentation of the greenhouse gas measurements on experimental sites of continuous cover forestry, 2021.

Weber, J., King, J. A., Abraham, N. L., Grosvenor, D. P., Smith, C. J., Shin, Y. M., Lawrence, P., Roe, S., Beerling, D. J., and Martin, M. V.: Chemistry-albedo feedbacks offset up to a third of forestation's $CO_2$ removal benefits, Science, 383, 860-864, https://doi.org/10.1126/science.adg6196, 2024.